# REASONER: An Explainable Recommendation Dataset with Comprehensive Labeling Ground Truths

**Xu Chen**[1,2]**, Jingsen Zhang**[1,2,]*__, Lei Wang**[1,2,]*__, Quanyu Dai**[4,]†__, Zhenhua Dong**[4],
**Ruiming Tang**[4], **Rui Zhang**[5], **Li Chen**[3], **Wayne Xin Zhao**[1,2], **Ji-Rong Wen**[1,2]
[1]Beijing Key Laboratory of Big Data Management and Analysis Methods, Beijing, China
[2]Gaoling School of Artificial Intelligence, Renmin University of China, Beijing, China
[3]Department of Computer Science, Hong Kong Baptist University, Hong Kong
[4]Huawei Noah's Ark Lab, China      [5]www.ruizhang.info

## Abstract

Explainable recommendation has attracted much attention from the industry and academic communities. It has shown great potential to improve the recommendation persuasiveness, informativeness and user satisfaction. In the past few years, while a lot of promising explainable recommender models have been proposed, the datasets used to evaluate them still suffer from several limitations, for example, the explanation ground truths are not labeled by the real users, the explanations are mostly single-modal and around only one aspect. To bridge these gaps, in this paper, we build a new explainable recommendation dataset, which, to our knowledge, is the first contribution that provides a large amount of real user labeled multi-modal and multi-aspect explanation ground truths. In specific, we firstly develop a video recommendation platform, where a series of questions around the recommendation explainability are carefully designed. Then, we recruit about 3000 high-quality labelers with different backgrounds to use the system, and collect their behaviors and feedback to our questions. In this paper, we detail the construction process of our dataset and also provide extensive analysis on its characteristics. In addition, we develop a library, where many well-known explainable recommender models are implemented in a unified framework. Based on this library, we build several benchmarks for different explainable recommendation tasks. At last, we present many new opportunities brought by our dataset, which are expected to promote the field of explainable recommendation. Our dataset, library and the related documents have been released at https://reasoner2023.github.io/.

## 1   Introduction

Recommender system has become an indispensable component in a large amount of real-world applications, ranging from e-commence [32] and entertainment [46] to education [30] and health-caring [37]. Basically, recommender system bridges the content providers with the users. For the content providers, better recommendations can effectively expose the items which are more likely to be clicked or purchased by the users, and thus, achieve better business targets. For the users, an effective recommender model can help to quickly discover the items of interest, and thus, save the user time and alleviate the problem of information overloading. Recently, explainable recommendation has been recognized as an important research direction [36, 2]. Essentially, providing explanations for the recommended items can further enhance the utilities of the content providers and users. For

---

*Equal contribution.
†Corresponding author.

37th Conference on Neural Information Processing Systems (NeurIPS 2023) Track on Datasets and Benchmarks.

Table 1: Comparison between different explainable recommendation datasets. "Real-U" means that whether the ground truths are labeled by the real users. "Multi-E" indicates that whether the dataset has multi-aspect explanations. "App-S" represents that whether the dataset can be generally used in different studies. "Multi-M" shows that whether the dataset contains multi-modal explanations. In addition, we also present the number of labelers in each dataset.

| Reference | Real-U | Multi-E | App-S | Multi-M | # Labelers |
|---|---|---|---|---|---|
| Extra [20] | ✓ | ✗ | ✓ | ✗ | <100 |
| VECF [8] | ✗ | ✗ | ✓ | ✗ | <100 |
| AFRec [24] | ✗ | ✗ | ✗ | ✗ | <100 |
| Ex3 [40] | ✗ | ✗ | ✗ | ✗ | <100 |
| MTER [38] | ✗ | ✓ | ✗ | ✗ | <100 |
| JRASA [28] | ✓ | ✓ | ✗ | ✗ | 100-300 |
| UseFCE [29] | ✓ | ✓ | ✗ | ✗ | <100 |
| ArgEx [15] | ✓ | ✓ | ✗ | ✓ | 100-300 |
| REASONER | ✓ | ✓ | ✓ | ✓ | ≈ 3000 |

the content providers, appropriate explanations can better persuade the users to interact with the recommended items [11]. For the users, reasonable explanations can reveal more informative item features to facilitate user decisions [18].

In the past few years, while a lot of promising explainable recommender models have been proposed, the datasets used to evaluate them still suffer from several limitations: to begin with, many work evaluate the models by recruiting annotators to label the ground truths on existing public datasets [27]. However, the annotators are not the real users in the datasets. Thus, the labeled results may deviate from the real user behavior reasons. Then, previous work mostly evaluate the explanations from only one aspect, for example, "whether the explanation can better persuade the users" [33] or "whether the explanation can reveal more informative item features" [6, 41]. However, the recommendation explanations may serve for different purposes [1], thus, these datasets may fail to comprehensively evaluate the model explainability. At last, while recent years have witnessed a few pioneer studies on evaluating the explanations from multiple aspects [39, 1, 3], the labeled datasets are not large enough, which may fail to evaluate the models in a statistically sufficient manner.

To alleviate the above problems, in this paper, we propose to build a large real user labeled dataset, which can evaluate the recommendation explanations from multiple aspects. We call our dataset as **REASONER**, where **RE** indicates that the ground truths are labeled by the real users, **ASON** represents that our dataset can evaluate multi-aspect explanations, **ER** is short for explainable recommendation. While this dataset is important for the explainable recommendation domain, it is not easy to build it. To begin with, if we build our dataset by extending an existing one, then it is hard to find the real users in the original dataset to label the explanation ground truths. If we do not rely on an existing dataset, then how to obtain the user-item interactions can be a challenge. In addition, different from the other domains, where the labeling tasks are usually objective (*e.g.*, identifying the contents of an image), we have to collect user subjective data, which is more sensitive to the designs of the labeling tasks, for example, "how to ask each question" and "what are the orders of the questions". Without proper designs, the annotators may misunderstand the questions or can be interfered by the external factors, which may lower the labeling accuracy. Last but not least, how to control the dataset quality is also a challenge. Our dataset aims to collect user subjective preferences, it can be difficult to purify it by directly judging whether the labeled results are right or wrong, since different users may have various opinions. We need to find effective rules to filter the results according to their reasonableness.

To overcome the above challenges, we propose to build our dataset in a two-step manner. To begin with, we develop a video recommendation platform, and carefully design a series of questions around the recommendation explainability. For example, "which features are the reasons that you would like to watch this video?" and "which features are most informative for this video?". Then, we recruit about 3000 labelers to use our platform and answer the above questions. This makes that the labelers of the explanation ground truth are exactly the real users that produce the user-item interactions[3]. To control the quality of our dataset, we set many rules to check and filter the labeled results. For example, "whether the labeling time is too short?" and "whether the answers of the same

---

[3]In the following, we do not discriminate the users and labelers.

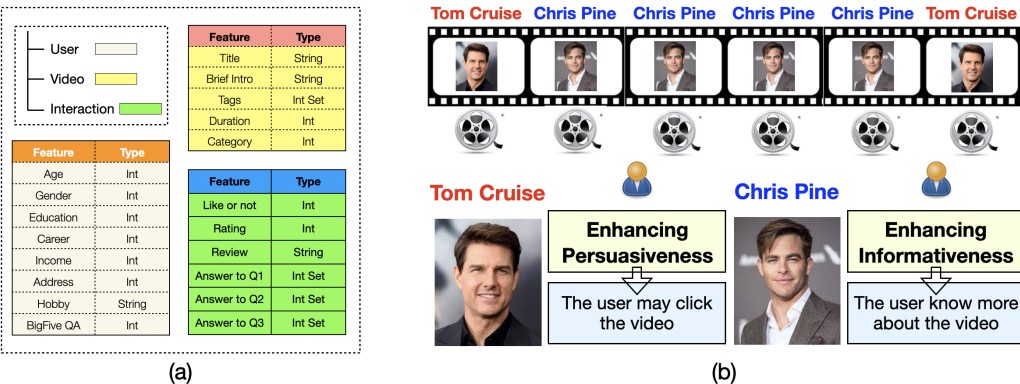

Figure 1: (a) All the information collected in our dataset. (b) Different ground truths for the recommendation explanations with different purposes.

user conflict with each other?". Beyond the above dataset, we also develop a library, which contains many well-known explainable recommender models. With this library, we build several benchmarks based on our dataset for different explainable recommendation tasks. Due to the space limitation, we present the details on the library and benchmarks in the appendix. At last, we point out several new opportunities brought by our dataset. We have released the above-mentioned dataset, library and documents at https://reasoner2023.github.io/ under a CC BY-NC 4.0 license.

## 2 Related Datasets

The datasets for evaluating explainable recommender models can be divided into two classes. In the first class, the datasets are usually generated by recruiting labelers to answer the questions around the model explainability. For example, [8] asks the labelers to imagine themselves as the real users and label the explanation ground truth. [24] requires the labelers to compare the explanations generated from the designed models and baselines. In [40, 22], the labelers have to evaluate the helpfulness of the explanations in terms of facilitating better user decisions. [35, 38] let the labelers to express their satisfactions and rate the informativeness of the explanations. [20] regards the user reviews as the explanation ground truths, and projects similar review sentences into the same cluster to facilitate model optimization. In this type of datasets, the labelers are usually not the real users, thus the labeled results may not accurately reflect the real user preference. In the second class, the explanation dataset is completely constructed by the labelers. For example, the authors in [28] recruit 286 labelers to evaluate the transparency and effectiveness of some given explainable recommender models. In [29], the authors employ 20 annotators to study the effectiveness of the feature-based explanations. In [15], 152 annotators are recruited, and the target is to study the user perception on the explanation types. While the results in these datasets can well represent the real user preference, the labeling questions are highly model-specific, which can be hardly used in the other studies. Comparing with the above datasets, REASONER has many significant advantages. For example, it is labeled by the real users and can be used to evaluate different models. In addition, it has multi-aspect explanation ground truths with different modalities. For clearly understanding the advantages of REASONER, we compare it with the previous explainable recommendation datasets in Table 1.

## 3 Dataset Construction

To align the labelers with the real users, our dataset is constructed based on two steps. The first step is building a recommendation platform, and designing many questions about the recommendation explainability. The second step is recruiting labelers to use the above platform and collecting their behaviors and answers to the above questions. In the following, we detail these two steps.

### 3.1 Building the Recommendation Platform

Comparing with traditional recommender systems, our platform is specially designed for collecting the user explainable behaviors. In this section, we firstly introduce the items on the platform and the

explanation candidates that the labelers can select from. Then, we describe the questions designed for collecting the explanation ground truths. Finally, we present the complete labeling process for each labeler based on our platform.

### 3.1.1 Items

We select videos as the recommendation items, since they are content intensive for providing sufficient explanation candidates. Considering that too long labeling time may distract the user attention, the length of the videos are confined to be shorter than three minutes. The selected videos can cover different categories, such as technology and music, which are expected to comprehensively represent the user preferences. All the videos are from an industrial short-video website.

### 3.1.2 Explanation candidates

We collect the explanation ground truths by asking the labelers to select from many video features, *e.g.*, "which features are the reasons that you would like to watch this video?". The most important features are:

● **Tags**. These are textual features, which are collected from three sources: (1) the original tags provided by the video authors. (2) The overall comments, which are posted by the viewers after watching the video. (3) The time-synchronized comments, which are provided by the viewers as they watching the video in a real-time manner [25]. The above information is from the same website as that of the videos. Since the overall and time-synchronized comments are originally raw texts, we process them by neural language processing tools [43] to extract the candidate tags. We combine all the tags from different sources, and manually remove the tags which are duplicated, meaningless or containing swear words.

● **Previews**. These are visual features, which are extracted from the videos. To make the previews more representative, we use a standard key frame extraction tool[4] for preview generation. We manually check the similarities between different previews, and remove the duplicated ones. For each video, we finally generate five previews for the labelers.

In addition to the tags and previews, we also present the video titles and brief introductions to the labelers. Besides, the labelers can also write the answers in a reserved text box, if they find that all the above features cannot represent their ideas.

### 3.1.3 Labeling questions

As mentioned before, the recommendation explanations may serve for different purposes. For the content provides, the explanations can be used to enhance the recommendation persuasiveness. For the users, the explanations can be leveraged to provide more informative item features for faster user decisions. As a result, there can be multiple ground truths for the explanations. For example, in Figure 1(b), the user is a fan of Tom Cruise. If we explain the video by "Tom Cruise", then the user may click the video, and the purpose of the content provided is achieved. From this perspective, "Tom Cruise" is the ground truth for improving the recommendation persuasiveness. However, while Tom Cruise is in the video, he only appears for a very short time, and the real leading actor is Chris Pine. If we explain the video by "Chris Pine", then the user can make more accurate and informed decisions. From this perspective, "Chris Pine" is the ground truth for enhancing the recommendation informativeness. To obtain the explanation ground truths from multiple aspects, we design a series of questions for the labelers to answer, and we introduce the most important ones as follows:

**Q1. Which features are the reasons that you would like to watch this video?** This question is asked before the user watches the video. The selected features are the user expectations, which reveal the reasons that motivate the users to watch the videos. This question aims to obtain the explanation ground truth in terms of enhancing the recommendation persuasiveness.

**Q2. Which features are most informative for this video?** This question is asked after the user has watched the video. At this time, the user has fully understood the video, thus the selected features can objectively represent the video. By this question, we would like to obtain the explanation ground truth in terms of enhancing the recommendation informativeness.

---

[4]https://github.com/FFmpeg/FFmpeg

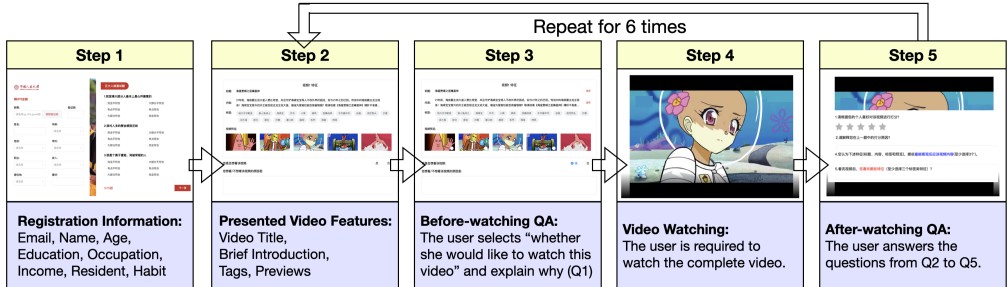

Figure 2: The complete labeling processing for each user to collect our dataset.

**Q3. Which features are you most satisfied with?** This question is also asked after the user has watched the video. It aims to obtain the explanation ground truth in terms of improving the user satisfaction. Comparing with Q1, this ground truth is obtained after the user has fully understood the video contents. Comparing with Q2, this ground truth involves the user subjective factors.

**Q4. Please give a rating (from 1 to 5) to this video according to your preference.** This question is designed to collect the user overall preference on the video.

**Q5. How do you comment this video?** This question aims to collect more detailed user preference on the video, and the user can express her opinions in a text box.

By the above questions, we collect the explanation ground truths from three aspects, that is, persuasiveness (Q1), informativeness (Q2) and user satisfactions (Q3). In summary, we present all the information collected in our dataset in Figure 1(a).

### 3.1.4 Workflow of the labeling task

Based on the above questions, in this section, we introduce the complete labeling process for each user (see Figure 2). In specific, there are five important steps:

**Step 1: User sign up**. To begin with, the users have to sign up our platform by providing their basic personal information. And then, we also require the users to complete a test on big five personality. This test includes 15 questions, which are standard and widely used over the world[5]. Based on the answers to these questions, we can understand the personality of a user from five dimensions, that is, openness, conscientiousness, extraversion, agreeableness and neuroticism. We refer the readers to [12, 13] for more detailed introduction on the personality test. The big five test may provide us with the opportunities to study the influence of different user personalities on the selection tendency of the recommendation explanations. For the above personal information, we have obtained the user permissions to use it for research purposes. To protect the user privacy, our dataset has been desensitized based on IDs.

**Step 2: Platform recommendation**. After the user has logged into the platform, the system will recommend her with three videos. For each video, the features introduced in Section 3.1.2 are also presented to the user. The videos are recommended in a completely random manner, that is, different videos are exposed to the user with the same probability. This setting aims to collect more diverse user preferences without the information cocoons effect. In addition, our dataset may also contribute to the field of debiased recommendation [7], where collecting uniform item exposure traffic can be too expensive in real-world scenarios [45].

**Step 3: User selection and before-watching question answering**. Once the user has received the recommendations, for each video, she has to carefully check its features, and decides whether she would like to watch this video. If the user decide to watch the video, then we require her to answer Q1 in Section 3.1.3, that is, "which features are the reasons that you would like to watch this video?". Otherwise, the user has to select the features which make her not want to watch the video.

**Step 4: Video watching**. In this step, the user has to watch the video. Since the video only lasts for a short time (<3min), the user can quickly understand it with little effort. For the videos that the user does not want to watch in the previous step, we also require her to watch them and answer the

---

[5]https://www.psytoolkit.org/survey-library/big5-bfi-s.html

questions in the following steps, which may help to study whether the videos of interests can be missed due to the improper explanations.

**Step 5: After-watching question answering**. After the video has been watched, we require the user to answer a series of questions. To begin with, the user has to provide her rating and comments on the video (corresponding to Q4 and Q5). Then, we require the user to select the features which are most informative for the videos (corresponding to Q2). At last, the user needs to select the features that they are most satisfied with (corresponding to Q3).

For each user, step 1 is only done for once, step 2 to step 5 are repeated for at least six times. Since the system recommends three videos every time, each user receives more than $3 \times 6$ (=18) recommendations in the complete labeling process.

## 3.2 Collecting the Dataset

Based on the above platform, we then recruit labelers to use it and complete the above labeling process. To control dataset quality, we design many rules to purify the labeled results.

### 3.2.1 Labelers

To guarantee the dataset quality, we cooperate with a professional company to recruit the labelers. For each labeler, we require that she should have been employed by the company before and have better working records. In addition, we need that the background of the labelers should be diverse enough. In total, we recruit about 3000 labelers, and we pay each labeler $8 USD for completing the labeling task. The overall cost of our dataset is about $28, 000 USD.

### 3.2.2 Dataset quality control

In traditional labeling tasks, such as the image recognition or entity tagging, the ground truths are objective, and it is easy to judge whether the labeled results are right or wrong, which makes it not hard to control the dataset quality. However, our dataset aims to collect the user subjective and personalized preferences. There is no strict right or wrong, which brings difficulties to control the dataset quality. To solve this problem, we firstly set a series of rules to judge whether the labeled results are reasonable, and then remove the unreasonable ones. In specific, the rules are mainly designed around the following aspects:

**A1. Labeling time.** On our platform, the time cost of the user for answering each question has been recorded in our database. For each question, we gather all the time costs of the users, and check whether there are outliers. For all the questions, if the same user is identified as an outlier for more than five times, then the user and all her labeled results are removed from the dataset.

**A2. Repeated answers.** For this aspect, we check whether the user answers to different questions are highly similar. For example, some labelers may always select the first three tags for Q1 across all the videos. Some labelers may use the same comment for different videos. In general, if a user has more than 1/3 similar answers, then we remove this user and her labeled results.

**A3. Conflict answers to the same question.** To check whether the labelers have carefully done our task, we intentionally insert many questions which have been answered by the labelers before. For the same labeler, we examine whether the answers to the same question are consistent. If a user has more than 1/3 inconsistent answers, then the user is removed.

**A4. Conflict answers across different questions.** Intuitively, if the labelers have carefully completed our task, then there should be some reasonable relations between different answers. For example, if the user give a very low rating, then the comments should be mostly negative. At first, we planned to summarize all the unreasonable relations and develop an automatic tool to identify them. However, we find that the relation patterns between different answers are too complex, and building a reliable enough tool may cost too much time. To solve these problems, we recruit five students in our lab to manually check each record in the dataset. To begin with, we require each student to independently check the dataset, and find out the unreasonable relations. Then, we merge all the unreasonable results, and let the students to vote for each one to determine whether it is reasonable or not. For the unreasonable results, we remove the corresponding records in the final datasets.

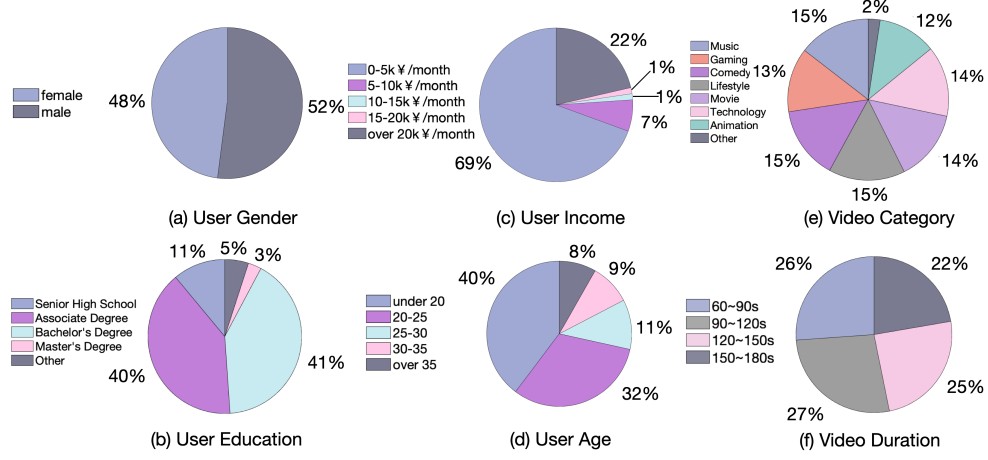

Figure 3: Statistics of the users and videos.

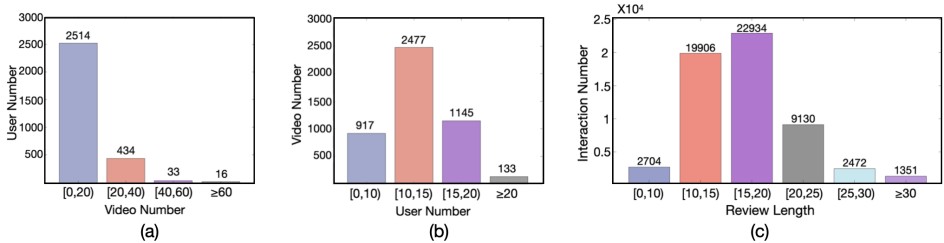

Figure 4: (a) The distribution of the number of users who watched videos in different ranges. (b) The distribution of the number of videos watched by users in different ranges. (c) The distribution of the user review length in words.

## 4 Statistical analysis

In this section, we introduce the basic statistics of our dataset, where we focus on the following aspects: (1) the basic statistics of the users and videos. (2) The statistics on the user-video, user-tag and video-tag interactions. (3) The similarities between the ground truths for different explanation aspects. (4) The statistics on the users who tend to select single- or multi-modality explanations. (5) The statistics on the length of the user reviews.

In our dataset, there are in total 2997 users, 4672 videos and 6115 tags. In Figure 3(a-d), we present the statistics of the users based on their genders, incomes, educations and ages. We can see the users can cover different populations, which makes our dataset more general and representative. In Figure 3(e-f), we present the statistics of the videos based on their categories and lengths. We can see, the categories of the videos are diverse, and for different categories, there are almost the same number of videos, which can help to explore more comprehensive and unbiased user preferences.

In Figure 4(a) and 4(b), we present the distribution of the user-video interactions. Each point $(x, y)$ represents that there are y users (or videos) who interact with $x$ videos (or users). For clear presentation, we merge the number of videos (or users) into four bins. In Figure 5, we show the distribution of the user-tag interactions. Since there are three types of tags for Q1, Q2 and Q3 as mentioned in Section 3.1.3, we present them separately. Each point $(x, y)$ in these figures represents that there are y users who totally select $x$ tags across different videos, where the x-axis is also clustered into four bins. In Figure 6, we show the distribution of the video-tag interactions. Similar to Figure 5, we present the results for Q1, Q2 and Q3 separately. Each point $(x, y)$ represents that there are $y$ videos whose tags have been totally selected for $x$ times by different users.

In our dataset, we label the ground truths for multi-aspect explanations. An interesting problem is that how different are the tags selected by a user for the same video on different explanation aspects. To answer this question, we measure the tag similarities based on the Jaccard distance. For every two explanation aspects, we firstly compute the tag similarity for each user-item pair, and then report the result by averaging all the user-video pairs in Table 2. We can see, for different explanation purposes,

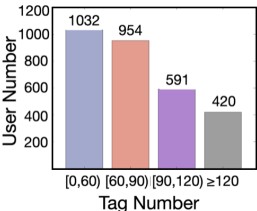 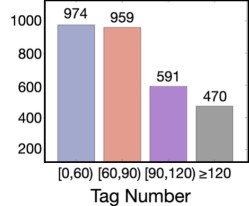 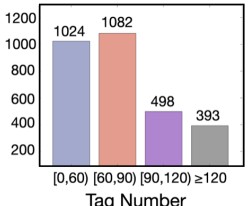

Figure 5: Distribution of the user-tag interactions. These three graphs respectively display the distribution of the number of users who label different numbers of Q1, Q2, and Q3 tags.

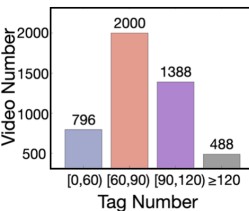 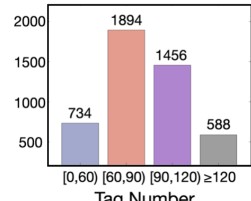 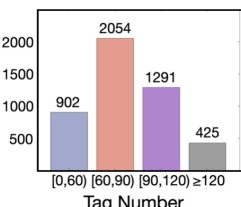

Figure 6: Distribution of the video-tag interactions. These three graphs respectively display the distribution of the number of videos labeled with different numbers of Q1, Q2, and Q3 tags.

the overlaps between the selected tags are usually not large (about 0.35 to 0.45). This demonstrates that one may need to use different tags to achieve different explanation purposes.

In our dataset, the users can select multi-modal explanations. Here, we compute the ratio between the numbers of users who select multi-modal explanations and all the users. We present the results for Q1, Q2 and Q3 separately in Table 3, where we can see about 38% users tend to select both textual and visual explanations, while the other users would like to select single-modal explanations. At last, we present the distribution of the user review lengths in Figure 4(c). We can see most of the reviews contain more than ten words, which can comprehensively reveal the user preferences.

Table 2: The average Jaccard similarity for different explanation aspects.

| Explanation aspects | Q1-Q2 | Q1-Q3 | Q2-Q3 |
|---|---|---|---|
| Jaccard similarity | 0.423 | 0.364 | 0.410 |

Table 3: Ratios between the number of users who select multi-modal explanations and the total number of users.

| Explanation aspect | Q1 | Q2 | Q3 |
|---|---|---|---|
| Ratio | 0.389 | 0.390 | 0.359 |

# 5 Library

Besides the above dataset, we also develop a library, where many well-known explainable recommender models are implemented in a unified framework. With this library, we build several benchmarks based on our dataset for different explainable recommendation tasks. Due to the space limitation, more details on the library and benchmarks are presented in the appendices.

## 5.1 The Structure of the Library

We show the structure of our library in Figure 7. The configuration module is the base part of the library and responsible for initializing all the parameters. We support three methods to specify the parameters, that is, the command line, parameter dictionary and configuration file. Based on the configuration module, there are four higher layer modules, that is,

**Data module**: this module converts the raw data into the model inputs. There are two components: the first one is responsible for loading the data and building vocabularies for the user reviews. The second part aims to process the data into the formats required by the model inputs, and generate the sample batches for model optimization.

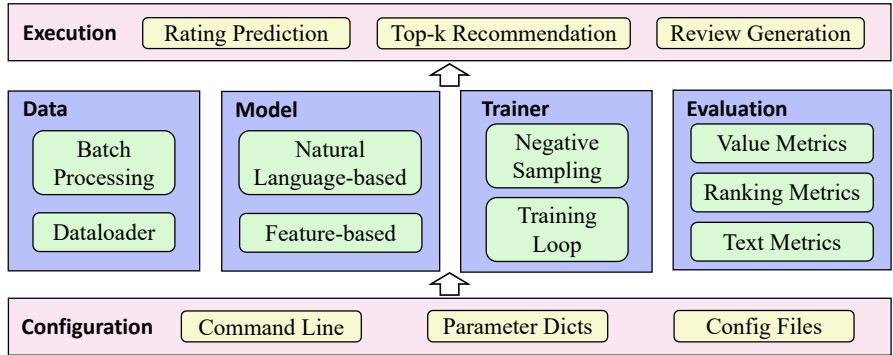

Figure 7: The structure of our library. There are six parts including the configuration, data, model, trainer, evaluation and execution modules.

**Model module**: this module implements the explainable recommender models. There are two types of methods in our library. The first one includes the feature-based explainable recommender models, and the second one contains the models with natural language explanations. We delay the detailed introduction of these models in the next section.

**Trainer module**: this module is leveraged to implement the training losses, such as the Bayesian Personalized Ranking (BPR) and Binary Cross Entropy (BCE). In addition, this module can also record the complete model training process.

**Evaluation module**: this module is designed to evaluate different models, and there are three types of evaluation tasks, that is, rating prediction, top-k recommendation and review generation.

Upon the above four modules, there is an execution module on the upper-most layer. It is responsible for optimizing models for different tasks, such as rating prediction and review generation.

### 5.2 Examples for Using Our Library

In this section, we introduce how to use our library. We present a simple example in Figure 8(a), where one can directly execute *tag_prediction.py* or *review_generate.py* to run a feature-based or review-based model, respectively. In each of these commands, one needs to specify three parameters to indicate the names of the model, dataset and configuration file, respectively.

Take *tag_prediction.py* for example, it sequentially executes the following steps: (1) Configuration. In this step, the parameters related to the model architecture and optimization process from different sources (commend line, configuration dictionary and files) are integrated into a dictionary. (2) Data loading. In this step, the dataloader is selected according to model type. For the review-aware models, this step reads all the records and build the vocabulary. (3) Data Formatting. The training, validation and test sets are processed into the formats required by the model input in a sample batch manner. (4) Initialization. The corresponding model class will be defined and initialized according to the parameter values in configuration. (5)-(6) Training. Selecting the optimization approach to train the model. (7) Evaluation. Measuring the model performance on different tasks.

Our library is highly extensible, and there are three steps to realize a new model: (1) implementing the basic functions of the model, including the model architecture, preference score prediction etc. (2) Customizing the training approaches in *train.py*. (3) Indicating the parameters in the config file.

## 6 New Opportunities

Based on the information collected in our dataset (see Figure 1(a)), we believe that our dataset can bring the following new opportunities for the domain of explainable recommendation:

• **Explainable recommendation with extensive personal information.** In most of the previous studies, the datasets used for training explainable recommender models do not contain sufficient personal information. By our dataset, people can access the desensitized user profiles, which may facilitate a lot research directions. For example, one can study the explanation fairness, where the sensitive variables can be the age, education, income and so on. In addition, people can also use the extensive personal information to enhance the predictive accuracy of the explanations.

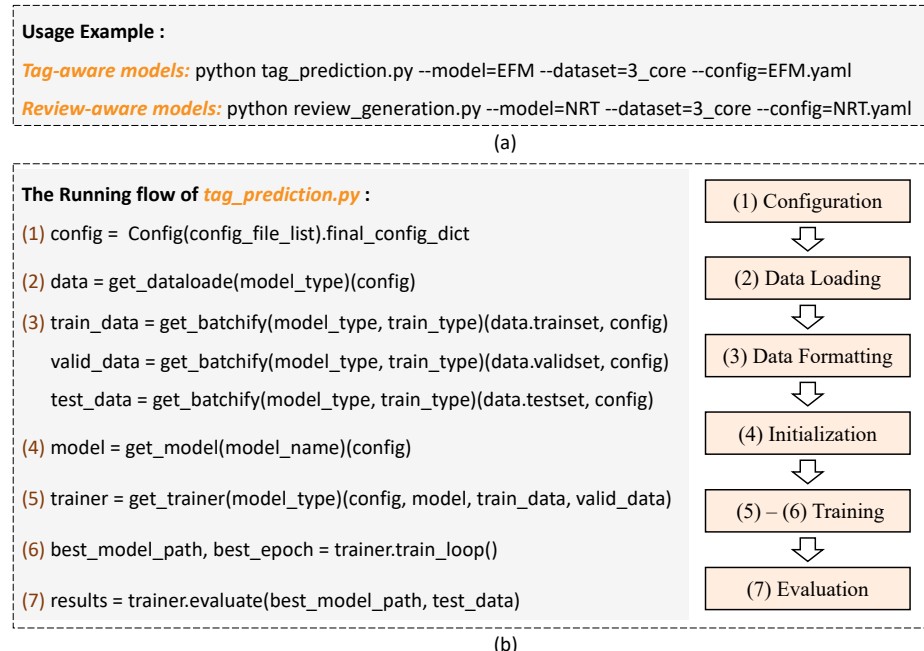

Figure 8: (a) Examples of the feature- and review-based models. (b) The running flow of our library.

• **Multi-aspect explainable recommendation.** For almost all the previous explainable recommender models, there is only one aspect ground truths for the explanations in the dataset. Thus, the learned models may not perform well on the other aspects. For example, if the model is only trained to improve the recommendation persuasiveness, then the generated explanations may not help to enhance (or even hurt) the recommendation informativeness. By our dataset, people can simultaneously consider different explanation aspects, and learn a more comprehensive explainable model to serve the on-line users. We argue that there can be much room in this direction, for example, if different explanation purposes have contradictions, how to design models to better overcome such contradictions.

• **Multi-modal explainable recommendation.** To our knowledge, most previous explainable recommender models only consider single-modal explanations. However, in the real-world scenarios, the users always need to perceive multi-modal information. By our dataset, people can conduct a lot of studies around the multi-modal explanations, for example: (1) which parts of users would like to select textual or visual features? (2) How to build a unified model to generate both textual and visual explanations? (3) What is the relation between the user personalities and the explanation modalities? And how to build a model to predict the explanation modality given a user personal information?

Beyond the above research opportunities on explainable recommendation, our dataset can also contribute the other research domains. For example, since our dataset is collected with uniform item exposure probabilities, people can use our dataset to study debiased recommendation. In our dataset, we have collected user big five personality information, thus, one can use it to investigate psychology-informed recommendation.

## 7   Conclusion and Future Work

In this paper, we introduced a new explainable recommendation dataset, where we detailed its construction process and extensively analyzed its characters. Our dataset can support multi-modal explanations, and the ground truths annotators are exactly the real users who produce the interactions. In addition, our dataset support multi-aspect explanations, and we have about 3000 users in the dataset. We developed an explainable recommendation toolkit, and built several benchmarks based on the above dataset for different explainable recommendation tasks. In addition, we also analyze the new opportunities brought by our dataset. Actually, our dataset is only a small step towards more measurable explainable recommendation, and there is still a long way to go. In the future, we plan to extend our dataset to the other domains, such as news, music and so on. In addition, we would also like employ more labelers to enlarge our dataset.

## Acknowledgments and Disclosure of Funding

This work is supported in part by National Key R&D Program of China (2022ZD0120103), National Natural Science Foundation of China (No. 62102420), Beijing Outstanding Young Scientist Program NO. BJJWZYJH012019100020098, Intelligent Social Governance Platform, Major Innovation & Planning Interdisciplinary Platform for the "Double-First Class" Initiative, Renmin University of China, Public Computing Cloud, Renmin University of China, fund for building world-class universities (disciplines) of Renmin University of China, Intelligent Social Governance Platform. The work is sponsored by Huawei Innovation Research Programs. We gratefully acknowledge the support from Mindspore[6], CANN (Compute Architecture for Neural Networks) and Ascend Ai Processor used for this research. We sincerely thank the contributions of Zhenlei Wang, Rui Zhou, Kun Lin, Zeyu Zhang, Jiakai Tang and Hao Yang from Renmin University of China for checking the unreasonable results in the dataset construction process.

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

# A  The Implemented Models in library

In our library, we implement two types of explainable recommender models, which are widely studied in the research community. The first one are feature-based explainable recommender models, where the features can be the tags, item aspects and so on. The second one are the models with natural language explanations.

More specifically, we implement the following representative feature-based explainable recommender models:

**EFM** [42] predicts the user preferences and generates explainable recommendations based on explicit product features and user opinions from the review information.

**TriRank** [14] models the user-item-aspect ternary relation as a heterogeneous tripartite graph based on user ratings and reviews, and it devises a vertex ranking algorithm for recommendation.

**LRPPM** [9] is a tensor-matrix factorization algorithm which captures the user preferences using ranking-based optimization objective over various item aspects.

**SULM** [4] enhances recommendations by recommending not only item but also the specific aspects by using aspect-level sentiment analysis.

**MTER** [38] is a tensor factorization method which models the task of item recommendation using a three-way tensor over the users, items and features. We omit the modeling of the opinions in the original implementation for adapting our data.

**AMF** [16] improves the recommendation accuracy by using the auxiliary information extracted from the user review aspects.

**TRDM** [48] introduces a two-stage approach to generate accurate item recommendations and effective tag-based potential features simultaneously for enhancing recommendation accuracy and diversity.

**TRAL** [47] proposes attention-based learning to capture diverse tag-based features, and compress these features with an attention pooling layer to enhance recommendation accuracy.

**HPTR** [44] employs hyperbolic distance to measure semantic relevance between entities, which better captures hierarchical structures presented in tag information.

**AIRec** [5] enhances tag-aware recommender system by employing a hierarchical attention network to capture multi-aspect preferences and leveraging tag intersection to improve conjunct feature learning.

**HAN-TR** [34] captures distinct user preferences and informative elements by employing separate attention networks for element-level influence and information-level attentiveness.

**TNAM** [17] addresses the issues of tag weight assignment in recommender systems by introducing a tag-based neural attention network that captures users' specific tag attention.

**BPR-T** [19] addresses high dimension and sparsity issues of tagging information by integrating tag mapping into a Bayesian personalized ranking collaborative filtering model.

In addition to the above shallow models based on matrix factorization, we also implement the following deep feature-based explainable recommender models (called **DERM** for short):

**DERM-MLP** is a deep recommender model for jointly predicting the ratings and tags. The two tasks share the set of user/item/tag embeddings. The hidden states as well as the tag embeddings are put into different layers corresponding to the different tasks.

**DERM-MF** firstly obtains a hidden state based on the user/item embeddings using matrix factorization, and then the outputs are computed by a neural network.

**DERM-C** combines matrix factorization and Multi-Layer Perceptron (MLP) to derive the hidden states, and the outputs are merged in a concatenated manner.

**DERM-H** leverages the tags to profile the users and items, and then use the same architecture as DERM-MLP for predicting the ratings and tags.

For the models with natural language explanations, we implement the following representative methods:

Table 4: Statistics of the datasets.

| Dataset | REASONER |
|---|---|
| # Users | 2,997 |
| # Items | 4,672 |
| # Tags | 6,115 |
| # Interactions | 58,497 |
| Avg. # words / review | 17 |

**Att2Seq** [10] is a review generation model which uses LSTM as the decoder, and output the texts directly based on the user/item IDs and rating information.

**NRT** [23] simultaneously predicts the reviews and ratings based on the input user-item pair, where the two tasks share the same embedding and hidden layers.

**PETER** [21] leverages Transformer to generate the user reviews, which is a state-of-the-art review generation model.

# B   Benchmark

## B.1   Experiment Setup

Considering that we have three types of ground truths for the explanations, we evaluate the model performance by predicting the tags for Q1, Q2, Q3, Q1+Q2, Q2+Q3, Q1+Q3 and Q1+Q2+Q3, respectively. When we have to predict multiple types of ground truths, we extend the original models to their multi-task versions by sharing the embedding parameters. We randomly split the dataset into the training, validation and testing sets according to the ratio of 8:1:1. For the review generation task, we use the most 20,000 frequently mentioned words to construct the vocabulary, and the maximum length of the generated sentences is set to 17, which is equal to the average length of reviews in dataset. The dataset statistics are presented in Table 4. For all the models, the batch size is set as 256. We tune the other key hyper-parameters by grid search. In specific, we tune the learning rate and the weight of L2 regularization in the range of [0.1, 0.01, 0.001, 0.0001] and [0.001, 0.0001, 0] respectively. For the deep models, we tune the hidden size and the layer number in the range of [32, 64, 128, 256] and [1, 2, 3, 4] respectively. More details of the experiment setting are shown in our project, which has been released at https://reasoner2023.github.io/. We use RMSE and MAE as the metrics to evaluate the performance of the rating prediction task. For the task of tag prediction, F1 and NDCG are selected to evaluate the model performance. To evaluate the quality of the generated reviews, we leverage the metrics including BLEU [31] and ROUGE [26] for model comparison.

## B.2   Experiment Results

The comparison results of the feature-based explainable recommender models on the tasks of tag and rating predictions are presented in Table 5-11. The comparison results of the models with natural language explanations on the task of review generation and rating prediction are presented in Table 12. We use the tags with top 10 prediction scores to calculate F1 and NDCG, and the results are percentage values with "%" omitted. For RMSE and MAE, a lower value indicates better performance. For each evaluation metric, we use bold fonts to label the best performance. Since the TriRank we implemented does not support to predict multiple types of tags simultaneously, we omit it in the corresponding tables.

Table 5: The benchmarking results of the feature-based explainable recommender models on predicting the tags for persuasiveness and ratings.

| Metrics | Persuasiveness | | Rating Prediction | |
| --- | --- | --- | --- | --- |
| | F1 | NDCG | RMSE | MAE |
| EFM | $26.99_{\pm0.35}$ | $18.89_{\pm0.23}$ | $1.68_{\pm0.00}$ | $1.24_{\pm0.01}$ |
| TriRank | $18.36_{\pm0.07}$ | $13.98_{\pm0.06}$ | $2.90_{\pm0.00}$ | $2.58_{\pm0.00}$ |
| LRPPM | $37.31_{\pm0.23}$ | $23.25_{\pm0.10}$ | $1.22_{\pm0.00}$ | $0.96_{\pm0.00}$ |
| SULM | $\mathbf{41.68}_{\pm0.63}$ | $\mathbf{25.77}_{\pm0.22}$ | $1.65_{\pm0.08}$ | $1.30_{\pm0.06}$ |
| MTER | $5.66_{\pm1.74}$ | $2.65_{\pm1.13}$ | $2.27_{\pm0.64}$ | $1.96_{\pm0.62}$ |
| AMF | $27.93_{\pm0.08}$ | $17.62_{\pm0.17}$ | $2.28_{\pm0.00}$ | $1.86_{\pm0.00}$ |
| DERM-MLP | $37.74_{\pm0.12}$ | $23.45_{\pm0.02}$ | $1.30_{\pm0.01}$ | $1.06_{\pm0.01}$ |
| DERM-MF | $36.57_{\pm0.14}$ | $21.49_{\pm0.17}$ | $1.32_{\pm0.00}$ | $1.14_{\pm0.00}$ |
| DERM-C | $37.17_{\pm0.19}$ | $23.21_{\pm0.01}$ | $1.30_{\pm0.00}$ | $1.07_{\pm0.01}$ |
| DERM-H | $35.58_{\pm0.07}$ | $22.02_{\pm0.18}$ | $1.27_{\pm0.01}$ | $1.04_{\pm0.01}$ |
| TRDM | $30.24_{\pm1.57}$ | $13.44_{\pm1.11}$ | $\mathbf{1.19}_{\pm0.00}$ | $\mathbf{0.95}_{\pm0.01}$ |
| TRAL | $5.93_{\pm0.05}$ | $2.61_{\pm0.02}$ | $1.29_{\pm0.00}$ | $1.08_{\pm0.00}$ |
| HPTR | $38.61_{\pm0.20}$ | $23.05_{\pm1.28}$ | $2.16_{\pm0.71}$ | $1.88_{\pm0.68}$ |
| AIRec | $38.05_{\pm0.07}$ | $23.07_{\pm0.03}$ | $1.31_{\pm0.00}$ | $1.08_{\pm0.01}$ |
| HAN-TR | $34.96_{\pm0.60}$ | $18.58_{\pm3.37}$ | $2.10_{\pm0.80}$ | $1.83_{\pm0.75}$ |
| TNAM | $5.97_{\pm0.01}$ | $2.60_{\pm0.01}$ | $1.37_{\pm0.01}$ | $1.17_{\pm0.01}$ |
| BPR-T | $32.70_{\pm0.62}$ | $18.32_{\pm0.82}$ | $1.23_{\pm0.00}$ | $0.98_{\pm0.01}$ |

Table 6: The benchmarking results of the feature-based explainable recommender models on predicting the tags for informativeness and ratings.

| Metrics | Informativeness | | Rating Prediction | |
| --- | --- | --- | --- | --- |
| | F1 | NDCG | RMSE | MAE |
| EFM | $5.38_{\pm0.28}$ | $3.97_{\pm0.19}$ | $1.68_{\pm0.00}$ | $1.24_{\pm0.01}$ |
| TriRank | $18.78_{\pm0.10}$ | $14.50_{\pm0.09}$ | $2.90_{\pm0.00}$ | $2.58_{\pm0.00}$ |
| LRPPM | $37.85_{\pm0.22}$ | $38.35_{\pm0.15}$ | $1.22_{\pm0.00}$ | $0.96_{\pm0.00}$ |
| SULM | $\mathbf{43.25}_{\pm0.59}$ | $\mathbf{42.97}_{\pm0.40}$ | $1.65_{\pm0.08}$ | $1.30_{\pm0.06}$ |
| MTER | $8.40_{\pm0.86}$ | $6.13_{\pm0.68}$ | $2.04_{\pm0.68}$ | $1.74_{\pm0.66}$ |
| AMF | $28.63_{\pm0.24}$ | $28.95_{\pm0.29}$ | $2.28_{\pm0.00}$ | $1.86_{\pm0.00}$ |
| DERM-MLP | $38.60_{\pm0.06}$ | $38.99_{\pm0.03}$ | $1.30_{\pm0.01}$ | $1.06_{\pm0.01}$ |
| DERM-MF | $37.10_{\pm0.09}$ | $35.37_{\pm0.14}$ | $1.32_{\pm0.00}$ | $1.14_{\pm0.00}$ |
| DERM-C | $37.96_{\pm0.10}$ | $38.43_{\pm0.05}$ | $1.30_{\pm0.00}$ | $1.07_{\pm0.01}$ |
| DERM-H | $36.36_{\pm0.55}$ | $35.85_{\pm0.41}$ | $1.29_{\pm0.01}$ | $1.06_{\pm0.01}$ |
| TRDM | $31.49_{\pm2.84}$ | $24.09_{\pm2.80}$ | $\mathbf{1.19}_{\pm0.00}$ | $\mathbf{0.95}_{\pm0.01}$ |
| TRAL | $6.04_{\pm0.06}$ | $4.59_{\pm0.01}$ | $1.28_{\pm0.00}$ | $1.08_{\pm0.00}$ |
| HPTR | $39.55_{\pm0.09}$ | $37.73_{\pm1.88}$ | $1.27_{\pm0.07}$ | $1.05_{\pm0.09}$ |
| AIRec | $38.90_{\pm0.06}$ | $39.33_{\pm0.06}$ | $1.31_{\pm0.00}$ | $1.08_{\pm0.01}$ |
| HAN-TR | $34.95_{\pm1.70}$ | $30.69_{\pm5.89}$ | $2.10_{\pm0.79}$ | $1.83_{\pm0.75}$ |
| TNAM | $37.77_{\pm0.35}$ | $37.97_{\pm0.20}$ | $1.36_{\pm0.00}$ | $1.16_{\pm0.00}$ |
| BPR-T | $33.83_{\pm0.43}$ | $31.15_{\pm0.33}$ | $1.23_{\pm0.01}$ | $0.98_{\pm0.01}$ |

Table 7: The benchmarking results of the feature-based explainable recommender models on predicting the tags for satisfaction and ratings.

| Metrics | Satisfaction | | Rating Prediction | |
|---|---|---|---|---|
| | F1 | NDCG | RMSE | MAE |
| EFM | $4.57_{\pm0.46}$ | $1.79_{\pm0.19}$ | $1.68_{\pm0.00}$ | $1.24_{\pm0.01}$ |
| TriRank | $16.82_{\pm0.03}$ | $13.16_{\pm0.02}$ | $2.90_{\pm0.00}$ | $2.58_{\pm0.00}$ |
| LRPPM | $36.04_{\pm0.16}$ | $22.35_{\pm0.08}$ | $1.22_{\pm0.00}$ | $0.96_{\pm0.00}$ |
| SULM | $\mathbf{40.46}_{\pm0.62}$ | $\mathbf{24.80}_{\pm0.19}$ | $1.64_{\pm0.09}$ | $1.29_{\pm0.07}$ |
| MTER | $5.97_{\pm1.92}$ | $2.85_{\pm1.07}$ | $2.26_{\pm0.65}$ | $1.96_{\pm0.62}$ |
| AMF | $27.16_{\pm0.19}$ | $17.05_{\pm0.21}$ | $2.28_{\pm0.00}$ | $1.86_{\pm0.00}$ |
| DERM-MLP | $36.76_{\pm0.07}$ | $22.53_{\pm0.12}$ | $1.30_{\pm0.01}$ | $1.06_{\pm0.01}$ |
| DERM-MF | $35.40_{\pm0.23}$ | $20.59_{\pm0.35}$ | $1.32_{\pm0.00}$ | $1.14_{\pm0.00}$ |
| DERM-C | $36.20_{\pm0.28}$ | $22.28_{\pm0.22}$ | $1.29_{\pm0.01}$ | $1.07_{\pm0.01}$ |
| DERM-H | $34.65_{\pm0.43}$ | $21.33_{\pm0.47}$ | $1.28_{\pm0.01}$ | $1.05_{\pm0.02}$ |
| TRDM | $31.29_{\pm0.63}$ | $14.74_{\pm0.69}$ | $\mathbf{1.19}_{\pm0.00}$ | $\mathbf{0.95}_{\pm0.01}$ |
| TRAL | $5.89_{\pm0.05}$ | $2.52_{\pm0.01}$ | $1.29_{\pm0.00}$ | $1.08_{\pm0.00}$ |
| HPTR | $35.35_{\pm2.90}$ | $17.78_{\pm3.92}$ | $1.81_{\pm0.77}$ | $1.56_{\pm0.72}$ |
| AIRec | $37.10_{\pm0.09}$ | $22.86_{\pm0.08}$ | $1.30_{\pm0.01}$ | $1.08_{\pm0.01}$ |
| HAN-TR | $33.95_{\pm1.80}$ | $18.06_{\pm4.15}$ | $2.10_{\pm0.80}$ | $1.83_{\pm0.75}$ |
| TNAM | $5.89_{\pm0.05}$ | $2.52_{\pm0.01}$ | $1.37_{\pm0.00}$ | $1.17_{\pm0.00}$ |
| BPR-T | $33.82_{\pm0.29}$ | $19.52_{\pm0.25}$ | $1.23_{\pm0.00}$ | $0.98_{\pm0.00}$ |

Table 8: The benchmarking results of the feature-based explainable recommender models on jointly predicting the tags for persuasiveness, informativeness and ratings.

| Metrics | Persuasiveness | | Informativeness | | Rating | |
|---|---|---|---|---|---|---|
| | F1 | NDCG | F1 | NDCG | RMSE | MAE |
| EFM | $15.69_{\pm0.03}$ | $12.74_{\pm0.07}$ | $5.38_{\pm0.73}$ | $3.94_{\pm0.42}$ | $1.66_{\pm0.00}$ | $1.23_{\pm0.00}$ |
| LRPPM | $37.32_{\pm0.21}$ | $23.26_{\pm0.09}$ | $37.89_{\pm0.19}$ | $38.37_{\pm0.13}$ | $1.22_{\pm0.00}$ | $0.96_{\pm0.00}$ |
| SULM | $\mathbf{41.34}_{\pm0.53}$ | $\mathbf{25.68}_{\pm0.20}$ | $\mathbf{42.70}_{\pm0.50}$ | $\mathbf{42.82}_{\pm0.35}$ | $1.67_{\pm0.06}$ | $1.31_{\pm0.05}$ |
| MTER | $35.53_{\pm0.21}$ | $21.88_{\pm0.35}$ | $36.22_{\pm0.30}$ | $36.09_{\pm0.61}$ | $1.36_{\pm0.01}$ | $1.09_{\pm0.01}$ |
| AMF | $27.67_{\pm0.16}$ | $17.45_{\pm0.15}$ | $28.23_{\pm0.33}$ | $28.57_{\pm0.34}$ | $2.28_{\pm0.00}$ | $1.86_{\pm0.00}$ |
| DERM-MLP | $38.49_{\pm0.15}$ | $23.80_{\pm0.10}$ | $39.14_{\pm0.10}$ | $39.38_{\pm0.10}$ | $1.30_{\pm0.01}$ | $1.06_{\pm0.01}$ |
| DERM-MF | $36.84_{\pm0.04}$ | $22.55_{\pm0.05}$ | $37.58_{\pm0.11}$ | $37.44_{\pm0.08}$ | $1.32_{\pm0.00}$ | $1.15_{\pm0.00}$ |
| DERM-C | $37.85_{\pm0.26}$ | $23.43_{\pm0.21}$ | $38.82_{\pm0.06}$ | $39.10_{\pm0.09}$ | $1.30_{\pm0.01}$ | $1.08_{\pm0.01}$ |
| DERM-H | $37.47_{\pm0.26}$ | $23.23_{\pm0.22}$ | $38.17_{\pm0.22}$ | $38.32_{\pm0.36}$ | $1.28_{\pm0.00}$ | $1.04_{\pm0.01}$ |
| TRDM | $32.91_{\pm1.06}$ | $15.51_{\pm0.72}$ | $33.01_{\pm0.50}$ | $25.02_{\pm0.37}$ | $\mathbf{1.19}_{\pm0.00}$ | $\mathbf{0.94}_{\pm0.01}$ |
| TRAL | $5.93_{\pm0.05}$ | $2.61_{\pm0.02}$ | $6.06_{\pm0.06}$ | $4.60_{\pm0.02}$ | $1.29_{\pm0.00}$ | $1.08_{\pm0.00}$ |
| HPTR | $38.68_{\pm0.25}$ | $23.11_{\pm1.20}$ | $36.20_{\pm4.79}$ | $31.40_{\pm9.27}$ | $2.16_{\pm0.71}$ | $1.88_{\pm0.68}$ |
| AIRec | $38.73_{\pm0.10}$ | $23.97_{\pm0.07}$ | $39.43_{\pm0.08}$ | $39.66_{\pm0.09}$ | $1.31_{\pm0.00}$ | $1.08_{\pm0.01}$ |
| HAN-TR | $33.32_{\pm0.75}$ | $16.83_{\pm0.53}$ | $33.31_{\pm0.93}$ | $27.72_{\pm0.46}$ | $1.29_{\pm0.02}$ | $1.06_{\pm0.03}$ |
| TNAM | $5.91_{\pm0.05}$ | $2.63_{\pm0.02}$ | $37.47_{\pm0.46}$ | $37.35_{\pm0.58}$ | $1.36_{\pm0.01}$ | $1.17_{\pm0.01}$ |
| BPR-T | $33.15_{\pm0.22}$ | $18.90_{\pm0.44}$ | $34.05_{\pm0.20}$ | $31.51_{\pm0.26}$ | $1.24_{\pm0.01}$ | $0.99_{\pm0.01}$ |

Table 9: The benchmarking results of the feature-based explainable recommender models on jointly predicting the tags for persuasiveness, satisfaction and ratings.

| Metrics | Persuasiveness | | Satisfaction | | Rating | |
|---|---|---|---|---|---|---|
| | F1 | NDCG | F1 | NDCG | RMSE | MAE |
| EFM | $15.58_{\pm0.03}$ | $12.84_{\pm0.07}$ | $4.58_{\pm0.45}$ | $1.77_{\pm0.18}$ | $1.66_{\pm0.00}$ | $1.23_{\pm0.00}$ |
| LRPPM | $37.32_{\pm0.21}$ | $23.26_{\pm0.09}$ | $36.06_{\pm0.14}$ | $22.37_{\pm0.07}$ | $1.22_{\pm0.00}$ | $0.96_{\pm0.00}$ |
| SULM | $\mathbf{41.36}_{\pm0.56}$ | $\mathbf{35.71}_{\pm0.22}$ | $\mathbf{40.15}_{\pm0.54}$ | $\mathbf{24.70}_{\pm0.18}$ | $1.67_{\pm0.06}$ | $1.31_{\pm0.05}$ |
| MTER | $5.83_{\pm0.52}$ | $2.56_{\pm0.18}$ | $5.36_{\pm0.17}$ | $2.23_{\pm0.12}$ | $2.04_{\pm0.68}$ | $1.74_{\pm0.66}$ |
| AMF | $27.71_{\pm0.21}$ | $17.52_{\pm0.21}$ | $26.90_{\pm0.18}$ | $16.94_{\pm0.16}$ | $2.28_{\pm0.00}$ | $1.86_{\pm0.00}$ |
| DERM-MLP | $38.37_{\pm0.09}$ | $23.71_{\pm0.12}$ | $37.32_{\pm0.02}$ | $22.87_{\pm0.03}$ | $1.30_{\pm0.01}$ | $1.06_{\pm0.01}$ |
| DERM-MF | $36.90_{\pm0.12}$ | $22.63_{\pm0.12}$ | $35.78_{\pm0.10}$ | $21.74_{\pm0.12}$ | $1.32_{\pm0.00}$ | $1.15_{\pm0.00}$ |
| DERM-C | $38.03_{\pm0.11}$ | $23.60_{\pm0.06}$ | $36.95_{\pm0.10}$ | $22.72_{\pm0.06}$ | $1.31_{\pm0.01}$ | $1.08_{\pm0.00}$ |
| DERM-H | $37.49_{\pm0.24}$ | $23.32_{\pm0.18}$ | $36.11_{\pm0.29}$ | $22.34_{\pm0.17}$ | $1.29_{\pm0.01}$ | $1.04_{\pm0.01}$ |
| TRDM | $32.57_{\pm1.92}$ | $15.11_{\pm1.21}$ | $30.91_{\pm1.77}$ | $13.86_{\pm1.40}$ | $\mathbf{1.19}_{\pm0.00}$ | $\mathbf{0.94}_{\pm0.00}$ |
| TRAL | $5.91_{\pm0.05}$ | $2.63_{\pm0.02}$ | $5.89_{\pm0.05}$ | $2.52_{\pm0.01}$ | $1.29_{\pm0.00}$ | $1.08_{\pm0.00}$ |
| HPTR | $38.64_{\pm0.20}$ | $22.93_{\pm1.46}$ | $32.96_{\pm4.03}$ | $14.78_{\pm4.06}$ | $2.16_{\pm0.71}$ | $1.88_{\pm0.68}$ |
| AIRec | $38.69_{\pm0.06}$ | $23.94_{\pm0.05}$ | $37.63_{\pm0.05}$ | $23.08_{\pm0.02}$ | $1.30_{\pm0.01}$ | $1.07_{\pm0.01}$ |
| HAN-TR | $35.95_{\pm2.15}$ | $19.75_{\pm3.93}$ | $34.95_{\pm2.02}$ | $19.13_{\pm3.74}$ | $2.09_{\pm0.80}$ | $1.83_{\pm0.75}$ |
| TNAM | $5.91_{\pm0.05}$ | $2.63_{\pm0.02}$ | $5.89_{\pm0.05}$ | $2.52_{\pm0.01}$ | $1.39_{\pm0.02}$ | $1.17_{\pm0.02}$ |
| BPR-T | $33.07_{\pm0.18}$ | $18.93_{\pm0.48}$ | $33.75_{\pm0.26}$ | $19.57_{\pm0.28}$ | $1.23_{\pm0.00}$ | $0.99_{\pm0.01}$ |

Table 10: The benchmarking results of the feature-based explainable recommender models on jointly predicting the tags for informativeness, satisfaction and ratings.

| Metrics | Informativeness | | Satisfaction | | Rating | |
|---|---|---|---|---|---|---|
| | F1 | NDCG | F1 | NDCG | RMSE | MAE |
| EFM | $5.15_{\pm0.79}$ | $3.75_{\pm0.38}$ | $4.57_{\pm0.54}$ | $1.75_{\pm0.21}$ | $1.66_{\pm0.00}$ | $1.23_{\pm0.00}$ |
| LRPPM | $37.89_{\pm0.19}$ | $38.37_{\pm0.14}$ | $36.06_{\pm0.14}$ | $22.37_{\pm0.07}$ | $1.22_{\pm0.00}$ | $0.96_{\pm0.00}$ |
| SULM | $\mathbf{42.70}_{\pm0.49}$ | $\mathbf{42.84}_{\pm0.37}$ | $\mathbf{40.12}_{\pm0.51}$ | $\mathbf{24.70}_{\pm0.17}$ | $1.67_{\pm0.06}$ | $1.31_{\pm0.05}$ |
| MTER | $6.13_{\pm0.03}$ | $4.56_{\pm0.18}$ | $5.64_{\pm0.66}$ | $2.37_{\pm0.30}$ | $2.04_{\pm0.68}$ | $1.74_{\pm0.66}$ |
| AMF | $28.17_{\pm0.28}$ | $28.53_{\pm0.32}$ | $26.79_{\pm0.06}$ | $16.89_{\pm0.09}$ | $2.28_{\pm0.00}$ | $1.86_{\pm0.00}$ |
| DERM-MLP | $39.23_{\pm0.08}$ | $39.45_{\pm0.02}$ | $37.40_{\pm0.07}$ | $22.93_{\pm0.06}$ | $1.30_{\pm0.01}$ | $1.06_{\pm0.01}$ |
| DERM-MF | $37.60_{\pm0.13}$ | $37.49_{\pm0.17}$ | $35.77_{\pm0.16}$ | $21.76_{\pm0.16}$ | $1.32_{\pm0.00}$ | $1.15_{\pm0.00}$ |
| DERM-C | $38.77_{\pm0.13}$ | $39.08_{\pm0.18}$ | $36.84_{\pm0.23}$ | $22.59_{\pm0.16}$ | $1.30_{\pm0.01}$ | $1.07_{\pm0.01}$ |
| DERM-H | $38.13_{\pm0.50}$ | $38.45_{\pm0.47}$ | $36.44_{\pm0.28}$ | $22.55_{\pm0.21}$ | $1.27_{\pm0.01}$ | $1.04_{\pm0.01}$ |
| TRDM | $33.15_{\pm0.98}$ | $25.24_{\pm1.92}$ | $30.42_{\pm1.50}$ | $13.83_{\pm1.22}$ | $\mathbf{1.19}_{\pm0.00}$ | $\mathbf{0.94}_{\pm0.01}$ |
| TRAL | $6.04_{\pm0.06}$ | $4.56_{\pm0.02}$ | $5.84_{\pm0.01}$ | $2.43_{\pm0.08}$ | $1.29_{\pm0.00}$ | $1.08_{\pm0.00}$ |
| HPTR | $37.36_{\pm3.15}$ | $32.22_{\pm7.55}$ | $35.35_{\pm2.93}$ | $17.79_{\pm3.92}$ | $1.81_{\pm0.77}$ | $1.56_{\pm0.72}$ |
| AIRec | $39.46_{\pm0.11}$ | $39.72_{\pm0.07}$ | $37.65_{\pm0.11}$ | $23.13_{\pm0.07}$ | $1.30_{\pm0.00}$ | $1.08_{\pm0.01}$ |
| HAN-TR | $35.79_{\pm3.09}$ | $32.11_{\pm6.85}$ | $34.04_{\pm2.94}$ | $18.48_{\pm4.06}$ | $1.31_{\pm0.01}$ | $1.10_{\pm0.02}$ |
| TNAM | $37.01_{\pm1.35}$ | $37.01_{\pm1.33}$ | $5.89_{\pm0.05}$ | $2.52_{\pm0.01}$ | $1.36_{\pm0.01}$ | $1.15_{\pm0.01}$ |
| BPR-T | $34.11_{\pm0.32}$ | $31.57_{\pm0.50}$ | $33.94_{\pm0.18}$ | $19.75_{\pm0.28}$ | $1.23_{\pm0.00}$ | $0.98_{\pm0.01}$ |

Table 11: The benchmarking results of the feature-based explainable recommender models on jointly predicting the tags for persuasiveness, informativeness and satisfaction and ratings.

| Metrics | Persuasiveness | | Informativeness | | Satisfaction | | Rating Prediction | |
|---|---|---|---|---|---|---|---|---|
| | F1 | NDCG | F1 | NDCG | F1 | NDCG | RMSE | MAE |
| EFM | $11.66_{\pm0.15}$ | $8.52_{\pm0.10}$ | $4.97_{\pm0.61}$ | $3.70_{\pm0.48}$ | $5.33_{\pm1.10}$ | $2.24_{\pm0.62}$ | $1.66_{\pm0.01}$ | $1.23_{\pm0.01}$ |
| LRPPM | $37.32_{\pm0.24}$ | $23.26_{\pm0.11}$ | $37.94_{\pm0.23}$ | $38.46_{\pm0.18}$ | $36.08_{\pm0.17}$ | $22.35_{\pm0.10}$ | $1.22_{\pm0.00}$ | $0.96_{\pm0.00}$ |
| SULM | $\mathbf{41.12}_{\pm0.50}$ | $\mathbf{25.35}_{\pm0.21}$ | $\mathbf{42.35}_{\pm0.45}$ | $\mathbf{42.66}_{\pm0.34}$ | $\mathbf{40.00}_{\pm0.49}$ | $\mathbf{24.66}_{\pm0.16}$ | $1.69_{\pm0.09}$ | $1.33_{\pm0.08}$ |
| MTER | $36.16_{\pm0.06}$ | $22.38_{\pm0.15}$ | $36.75_{\pm0.09}$ | $36.94_{\pm0.24}$ | $34.84_{\pm0.09}$ | $21.52_{\pm0.03}$ | $1.34_{\pm0.04}$ | $1.08_{\pm0.03}$ |
| AMF | $27.83_{\pm0.37}$ | $17.47_{\pm0.19}$ | $28.35_{\pm0.33}$ | $28.66_{\pm0.34}$ | $27.09_{\pm0.34}$ | $17.03_{\pm0.16}$ | $2.28_{\pm0.00}$ | $1.86_{\pm0.00}$ |
| DERM-MLP | $38.60_{\pm0.08}$ | $23.81_{\pm0.07}$ | $39.33_{\pm0.09}$ | $39.57_{\pm0.05}$ | $37.52_{\pm0.09}$ | $22.97_{\pm0.09}$ | $1.31_{\pm0.02}$ | $1.07_{\pm0.02}$ |
| DERM-MF | $37.42_{\pm0.21}$ | $23.16_{\pm0.08}$ | $38.26_{\pm0.13}$ | $38.46_{\pm0.16}$ | $36.60_{\pm1.01}$ | $22.18_{\pm0.19}$ | $1.33_{\pm0.00}$ | $1.16_{\pm0.00}$ |
| DERM-C | $38.05_{\pm0.22}$ | $23.53_{\pm0.07}$ | $39.03_{\pm0.15}$ | $39.29_{\pm0.11}$ | $37.19_{\pm0.15}$ | $22.79_{\pm0.08}$ | $1.30_{\pm0.01}$ | $1.08_{\pm0.01}$ |
| DERM-H | $37.64_{\pm0.24}$ | $23.36_{\pm0.18}$ | $38.52_{\pm0.44}$ | $38.83_{\pm0.39}$ | $36.70_{\pm0.40}$ | $22.60_{\pm0.16}$ | $1.28_{\pm0.01}$ | $1.05_{\pm0.02}$ |
| TRDM | $33.50_{\pm1.85}$ | $15.64_{\pm1.83}$ | $33.94_{\pm1.06}$ | $26.47_{\pm1.89}$ | $31.79_{\pm1.17}$ | $14.77_{\pm1.09}$ | $\mathbf{1.19}_{\pm0.00}$ | $\mathbf{0.94}_{\pm0.01}$ |
| TRAL | $5.88_{\pm0.14}$ | $2.56_{\pm0.04}$ | $5.91_{\pm0.09}$ | $4.48_{\pm0.17}$ | $35.42_{\pm0.20}$ | $21.09_{\pm0.08}$ | $1.29_{\pm0.00}$ | $1.07_{\pm0.00}$ |
| HPTR | $38.82_{\pm0.38}$ | $22.14_{\pm1.49}$ | $39.31_{\pm0.50}$ | $38.01_{\pm2.21}$ | $37.64_{\pm0.26}$ | $21.46_{\pm1.51}$ | $1.77_{\pm0.98}$ | $1.50_{\pm0.93}$ |
| AIRec | $38.89_{\pm0.09}$ | $23.98_{\pm0.05}$ | $39.61_{\pm0.11}$ | $39.82_{\pm0.09}$ | $37.85_{\pm0.05}$ | $23.16_{\pm0.06}$ | $1.30_{\pm0.00}$ | $1.07_{\pm0.01}$ |
| HAN-TR | $37.95_{\pm0.63}$ | $23.46_{\pm0.02}$ | $38.65_{\pm0.70}$ | $39.01_{\pm0.39}$ | $37.07_{\pm0.82}$ | $22.74_{\pm0.04}$ | $1.80_{\pm0.67}$ | $1.57_{\pm0.62}$ |
| TNAM | $37.96_{\pm0.13}$ | $23.51_{\pm0.02}$ | $38.74_{\pm0.23}$ | $38.81_{\pm0.17}$ | $5.89_{\pm0.08}$ | $2.44_{\pm0.12}$ | $1.37_{\pm0.01}$ | $1.17_{\pm0.01}$ |
| BPR-T | $33.41_{\pm0.44}$ | $19.39_{\pm0.48}$ | $34.37_{\pm0.28}$ | $32.15_{\pm0.36}$ | $34.29_{\pm0.15}$ | $20.11_{\pm0.22}$ | $1.24_{\pm0.01}$ | $1.00_{\pm0.02}$ |

Table 12: The benchmarking results of the models with natural language explanations in our library. For BLEU and ROUGE, the results are percentage values with "%" omitted. "-" means the evaluation metric is not available for the model.

| Metrics | BLEU (%) | | ROUGE-1 (%) | | | ROUGE-2 (%) | | |
|---|---|---|---|---|---|---|---|---|
| | B-1 | B-4 | F1 | R | P | F1 | R | P |
| Att2Seq | $\mathbf{19.96}_{\pm0.27}$ | $\mathbf{3.25}_{\pm0.19}$ | $\mathbf{22.13}_{\pm0.27}$ | $\mathbf{19.73}_{\pm0.44}$ | $26.40_{\pm0.78}$ | $\mathbf{5.56}_{\pm0.08}$ | $\mathbf{5.19}_{\pm0.16}$ | $6.26_{\pm0.24}$ |
| NRT | $17.67_{\pm1.10}$ | $2.92_{\pm0.65}$ | $20.60_{\pm0.57}$ | $16.04_{\pm1.27}$ | $\mathbf{30.02}_{\pm2.40}$ | $5.23_{\pm0.56}$ | $4.20_{\pm0.77}$ | $\mathbf{7.33}_{\pm0.42}$ |
| PETER | $17.65_{\pm1.18}$ | $2.35_{\pm0.35}$ | $20.00_{\pm1.07}$ | $15.68_{\pm1.62}$ | $28.59_{\pm1.42}$ | $4.99_{\pm0.48}$ | $3.95_{\pm0.58}$ | $7.00_{\pm0.57}$ |

