# OpenReview forum: "REASONER: An Explainable Recommendation Dataset with Comprehensive Labeling Ground Truths"
_NeurIPS.cc/2023/Track/Datasets_and_Benchmarks — NeurIPS 2023 Datasets and Benchmarks Poster_

### Official Review · Reviewer_ZxYM · 2023-07-14
**REASONER: An Explainable Recommendation Dataset with Comprehensive Labeling Ground Truths**

**Rating:** 6
**Confidence:** 5
**Clarity:** This paper is well organized.

**Strengths:**

1. The motivation behind this work is compelling as it addresses several limitations present in previous studies. These limitations include discrepancies between labeled results and actual user behavior, evaluations focused on a single aspect, and inadequate sizes of labeled datasets.
2. The contributions are substantial. This library is an invaluable asset for researchers and practitioners in the field, equipping them with a comprehensive toolkit to construct and assess explainable recommendation systems effectively.
3. The authors have conducted an extensive array of analyses, providing a thorough evaluation of the proposed dataset. These analyses showcase the dataset's performance comprehensively, offering insights into its strengths and capabilities.


**Additional Feedback:**

1.	Can the authors show some cases about multi-model explanations? These case examples would provide a clearer understanding of how the dataset facilitates multi-modal explanations
2.	The authors claim that the dataset can be generally used in different studies, while other datasets can not. Can the authors give a clearer explanation?


**Correctness:**

The claims made in the submission are correct. The authors construct the dataset based on two steps. The first step is building a recommendation platform, and designing many questions about the recommendation explainability. The second step is recruiting labelers to use the above platform and collecting their behaviors and answers to the above questions. These two steps are reasonable.

**Documentation:**

Clear collection process for the dataset and easy to use data.

**Ethics:**

No ethical concerns.

**Limitations:**

1. It is recommended that the authors compare the distribution gap between the 3,000 labelers and the real users to ensure the representativeness of the labeling process. By conducting such a comparison, the authors can assess how closely the labelers' behavior aligns with that of actual users, providing valuable insights into the dataset's reliability and validity.
2. Providing analysis and justification for the manual rules implemented in the dataset construction process would enhance the transparency and understanding of the dataset's quality. By presenting the rationale and empirical evidence supporting these rules, the authors can strengthen the confidence in the dataset and address any concerns regarding the rationality of the rules.
3. Conducting case studies to showcase the evaluation of multi-aspect explanations would be beneficial in demonstrating the dataset's capability. By selecting representative examples and thoroughly evaluating the explanations across multiple aspects, the authors can provide concrete evidence of the dataset's ability to handle and evaluate complex recommendation scenarios.


**Opportunities For Improvement:**

1. The dataset primarily focuses on videos; however, it has the potential to be extended to other domains, such as news, music, and more. Expanding the dataset to encompass diverse domains would enhance its applicability and make it more versatile for various recommendation tasks.
2. In order to ensure the accuracy of the explanation ground truth, the authors have employed approximately 3000 labelers who represent the real users responsible for generating the user-item interactions. While the use of a large number of labelers is beneficial, it is necessary to support the claim that this approach accurately reflects the distribution of real users through experimental data.
3. The authors have employed numerous artificial rules to guarantee the quality of the data; however, the rationale behind these rules lacks empirical evidence. It is important to provide data-supported justifications for the rules to enhance confidence in the dataset's quality and its ability to reflect real-world scenarios accurately.
4. The authors state that their dataset allows for the evaluation of multi-aspect explanations; however, this claim is not demonstrated explicitly clear. It would be beneficial for the authors to provide specific examples or results that illustrate the dataset's capability to evaluate explanations across multiple aspects, thereby strengthening their assertion.


**Relation To Prior Work:**

The authors compare their dataset with the previous explainable recommendation datasets in Table 1, which is clear.

**Summary And Contributions:**

The authors present a new explainable recommendation dataset that enables multi-modal and multi-aspect explanations. They provide a thorough description of how the dataset was constructed and conduct a comprehensive analysis of its features. Additionally, they develop a library that implements ten popular explainable recommender models within a unified framework. Using this library, they create benchmarks for various explainable recommendation tasks. The authors conclude by discussing the potential opportunities that the dataset brings to the field of explainable recommendation, anticipating its positive impact on advancing research in this area.

---

> ### Author Response · Authors · 2023-08-16
>
> Thanks for your comments. We try to alleviate your concerns in the following.
>
> (1) The dataset primarily focuses on videos; however, it has the potential to be extended to other domains, such as news, music, and more. Expanding the dataset to encompass diverse domains would enhance its applicability and make it more versatile for various recommendation tasks.
>
> Thanks for this comment, extending this dataset to the other domains is definitely our next step. We plan to recruit more labelers, or directly conduct online experiments to build more explainable datasets for various recommendation scenarios.
>
> (2) In order to ensure the accuracy of the explanation ground truth, the authors have employed approximately 3000 labelers who represent the real users responsible for generating the user-item interactions. While the use of a large number of labelers is beneficial, it is necessary to support the claim that this approach accurately reflects the distribution of real users through experimental data.
>
> Thank you for this comment. We believe in real-world scenarios, the user distribution may vary with the specific applications. In our dataset, we first try to ensure that our dataset contain similar numbers of male and female users, and then for the other demographic features, we try to make them as diverse as possible. If one need to study some specific applications, they can sample a small part from our dataset.
>
> (3) The authors have employed numerous artificial rules to guarantee the quality of the data; however, the rationale behind these rules lacks empirical evidence. It is important to provide data-supported justifications for the rules to enhance confidence in the dataset's quality and its ability to reflect real-world scenarios accurately.
>
> Thanks for your comment. These artificial rules have been primarily designed based on common understandings, making it difficult to evaluate their effectiveness. In order to ensure the credibility of the obtained results, some strict criteria have been established. For instance, if a user provides more than 1/3 repetitive answers, they are removed from the analysis. In reality, if a user is not diligent in taking our tests, most of their answers tend to be repetitive. Additionally, all labelers are compensated for their work, so if they fail to carefully label the questions, they may not receive their payment.
>
> (4) The authors state that their dataset allows for the evaluation of multi-aspect explanations; however, this claim is not demonstrated explicitly clear. It would be beneficial for the authors to provide specific examples or results that illustrate the dataset's capability to evaluate explanations across multiple aspects, thereby strengthening their assertion.
>
> Thanks for your comment. In the experiments detailed in the Appendix (see Table 4,5,6, and 7), we have conducted extensive experiments to evaluate the performances of different models in providing multi-aspect explanations.
>
> (5) Can the authors show some cases about multi-model explanations? These case examples would provide a clearer understanding of how the dataset facilitates multi-modal explanations
>
> Thank you for your comment. In the process of constructing the dataset, we gathered two modalities of video features as explanation candidates, namely textual features and visual features (see Figure 2). Specifically, textual features consist of video's title, introduction, and tags, and visual features are some representative previews of videos.
>
> To our knowledge, most previous explainable recommender models only consider single-modal explanations. However, in the real-world scenarios, the users always need to perceive multi-modal information. By our dataset, people can conduct a lot of studies around the multi-modal explanations.
>
> (6) The authors claim that the dataset can be generally used in different studies, while other datasets can not. Can the authors give a clearer explanation?
>
> Thank you for your comment. For instance, many previous datasets with explanations only focus on a single aspect. If our goal is to develop a recommender model capable of providing explanations from multiple perspectives, our dataset proves to be invaluable for conducting such study. The strengths of our dataset lie in its thorough labeling of the ground truth for explanations.

---

> > ### Author Response · Authors · 2023-08-18
> > **Thanks for the comments**
> >
> > Dear reviewer ZxYM,
> >
> > Thanks again for your significant comments, which can definitely improve our paper. In the rebuttal and re-submitted paper, we try to explain your questions one by one. In addition, we have added a large number of experiments to make our paper more solid. If you have further questions, we are very happy to discuss them.
> >
> > Thanks

---

> > > ### Author Response · Authors · 2023-08-29
> > > **Thanks for the comments**
> > >
> > > Dear Reviewer ZxYM,
> > >
> > > Thanks so much for your hard work in reading and reviewing our paper.
> > >
> > > Considering that the rebuttal ddl is approaching rapidly, we would like to kindly ask whether our rebuttal and additional experiments can alleviate your concerns. If you have more questions, we would like to grasp the last opportunities to discuss them for a higher score.
> > >
> > > Your input is highly valued and helps us improve our work.
> > >
> > > Thanks

---

### Official Review · Reviewer_pC4V · 2023-07-21
**review of REASONER**

**Rating:** 6
**Confidence:** 3
**Correctness:** Yes. The dataset seems to be construc…
**Clarity:** Yes. The paper is clear and well-writ…

**Strengths:**

1.The paper provides datasets labeled by real users with multi-aspect, multi-modal explanations, in favor of explainable recommendations, and recommendation fairness.

2.The construction of the dataset is clearly described and well-designed, and detailed documentation is provided.

3.Some interesting statistical analyses are provided and the dataset deserves to be further analyzed and explored.

**Additional Feedback:**

N/A

**Documentation:**

Yes. The dataset and library are well documented. The data collection, organization, and availability are well described.

**Ethics:**

No concerns. The dataset is desensitized to protect the user privacy.

**Limitations:**

I think there is no obvious limitation and potential negative societal impact of this work.

**Opportunities For Improvement:**

1.Recommendation datasets tend to be relatively large in size, but this one seems not so large, and I'm not sure it's enough for recommendation model training.

2.The labeled platform is implemented in Chinese (Figure 2), but the presented dataset is in English. If the dataset is translated from Chinese to English, I'm concerned about the quality of the dataset due to the translation.

The caption in Table 3 is not consistent with the description in Section 4 Line 272, please check them.

**Relation To Prior Work:**

Yes. The differences between this dataset and previous datasets are present in Table 1, which demonstrates that this dataset is labeled by the real users, and provides multi-aspect and multi-modal explanations.

**Summary And Contributions:**

This paper proposes to create an explainable recommendation dataset with multi-aspect and multi-modal explanations labeled by the real users. The authors develop a video recommendation platform and then recruit labelers to use it. Then the data is collected and processed to construct the dataset. Some statistical analyses are presented.

The authors also develop a library, which contains some baseline models, and benchmarking results are provided in the supplementary material.

---

> ### Author Response · Authors · 2023-08-16
>
> Thanks for your comments.
>
> (1) Recommendation datasets tend to be relatively large in size, but this one seems not so large, and I'm not sure it's enough for recommendation model training.
>
> Thanks for this comment. The meaning of this dataset is to make up the gap that most public datasets do not have explanation ground truth. In the field of explainable recommendation, our dataset is already 10 times larger than the ones which are usually leveraged to evaluate the explainable models (see table 1). While the scale of our dataset is not that large as the ones without explanation ground truth, it can provide many new opportunities which has been ignored by the existing datasets. We believe the contributions are non-trivial.
>
> In the next version, we will further increase the size of the dataset. We will try to use the currently labeled data as a seed set to design corresponding instructions and prompts, and use the language capabilities of the large language model to generate a larger-scale explainable recommendation dataset.
>
> (2) The labeled platform is implemented in Chinese (Figure 2), but the presented dataset is in English. If the dataset is translated from Chinese to English, I'm concerned about the quality of the dataset due to the translation.
>
> Thanks for this comment. For each translation, we have manually checked its correctness, which took a lot of effort to ensure the quality of dataset. And for the incorrect ones, we revise them based on human-beings.
>
> (3) The caption in Table 3 is not consistent with the description in Section 4 Line 272, please check them.
>
> Thanks for the comment. We have checked this error and revised it in the resubmitted version.

---

> > ### Author Response · Authors · 2023-08-18
> > **Thanks for the comments**
> >
> > Dear reviewer pC4V,
> >
> > Thanks again for your significant comments, which can definitely improve our paper. In the rebuttal and re-submitted paper, we try to explain your questions one by one. In addition, we have added a large number of experiments to make our paper more solid. If you have further questions, we are very happy to discuss them.
> >
> > Thanks

---

> > > ### Author Response · Authors · 2023-08-29
> > > **Thanks for the comments**
> > >
> > > Dear Reviewer pC4V,
> > >
> > > Thanks so much for your hard work in reading and reviewing our paper.
> > >
> > > Considering that the rebuttal ddl is approaching rapidly, we would like to kindly ask whether our rebuttal and additional experiments can alleviate your concerns. If you have more questions, we would like to grasp the last opportunities to discuss them for a higher score.
> > >
> > > Your input is highly valued and helps us improve our work.
> > >
> > > Thanks

---

### Official Review · Reviewer_1MRi · 2023-07-21
**A Significant Step Towards Better Explainability in Recommender Systems**

**Rating:** 6
**Confidence:** 5

**Strengths:**

1) The work offers a significant improvement over current datasets used for evaluating explainable recommendation models, addressing limitations such as absence of real-user labelling and limited aspects of explanation.
2) The authors' efforts to ensure data quality are noteworthy. Rules to check and filter labeled results have been set, considering factors like labeling time and the consistency of a user's responses.
3) The authors have made an extensive analysis of their dataset, including an in-depth presentation of its construction process. This transparency enhances the reproducibility of their work.
4) The work opens new opportunities for future studies in the field of explainable recommendation.

**Additional Feedback:**

It would be beneficial for the authors to discuss potential biases in the dataset due to the recruitment of labelers from various backgrounds. It may also be worth discussing how the approach might be generalized to other kinds of content beyond video.

**Clarity:**

The paper is well written, with a clear explanation of the problems addressed, the construction process of the dataset, and the proposed solutions. The information is organized logically and the authors provide adequate context.

**Correctness:**

The claims made in the submission seem to be correct based on the presented methodology and results. However, the addition of error bars to the results would improve confidence in the correctness of the experimental outcomes.

**Documentation:**

The dataset is well-documented with clear steps on data collection and organization. However, the authors should consider including a more detailed dataset sheet in the supplementary material.

**Ethics:**

They should also discuss any potential misuse of the dataset and how such issues could be mitigated.

**Limitations:**

1) The authors should explicitly discuss the potential limitations of the newly constructed dataset, including possible biases in the data collection process or inherent limitations due to the specific domain (i.e., video recommendation).
2) Consideration should also be given to addressing any potential negative societal impacts of this work. The authors may want to discuss how the dataset can be used responsibly to ensure fairness and privacy.
3) The authors should consider discussing how the subjectivity of explanation satisfaction could potentially influence the model's evaluation.
4) The dataset depends on subjective labeler feedback, which may introduce inherent biases and inconsistencies. Although the authors have taken measures to ensure data quality, the subjectivity remains a limitation.

**Opportunities For Improvement:**

1) Lack of Error Bars in Experiment Results: The authors have presented the results of their experiments, however, they have not included error bars in these results. This lack of reporting on the variability of the data could impede the replicability of the experiments, thereby undermining the robustness of the study. It would be beneficial for the authors to include error bars in their experiment results to enhance transparency and reproducibility.
2) Insufficient Detail in the Supplementary Material: The authors should consider including a dataset sheet in the supplementary material, as per standard recommendations. This sheet would provide more detailed information about the dataset, further enhancing the understanding and usability of the data.

**Relation To Prior Work:**

The work is well-positioned within existing literature. The authors clearly state how their work differs from previous studies and how it addresses the limitations of current datasets used for evaluating explainable recommender models.

**Summary And Contributions:**

The paper presents REASONER, an explainable recommendation dataset that fills in existing gaps in the recommendation field. The main contribution lies in the dataset's comprehensiveness, involving real user-labeled multi-modal, multi-aspect explanation ground truths. The authors developed a video recommendation platform, recruited labelers, and collected their feedback. Furthermore, the authors provide a library with well-known explainable recommender models and establish benchmarks for different tasks.

---

> ### Author Response · Authors · 2023-08-16
>
> Thanks for the comments. In the following, we try to alleviate your concern one by one.
>
> (1) Lack of Error Bars in Experiment Results: It would be beneficial to include error bars in experiment results.
>
> Thanks for this comments. In the following, **we have conducted further experiments to show the error bars of the results in the resubmitted Appendix**. It should be pointed out that the benchmark is not the focus of this paper, our key contribution is to propose the explainable recommendation dataset with comprehensive labeling ground truths.
>
> (2) Insufficient Detail in Supplementary Material: The authors should consider including a dataset sheet in supplementary material.
>
> Thanks for this comments. The detailed information about the dataset have been presented in the main paper. We have conducted very extensive analysis on the dataset characters in section 4. And the division method of dataset for experiments has been introduced in C.1 of the Appendix.  **Following your advice, we have incorporated the dataset information into the resubmitted appendix (see Table 1) to make it self-contained**.
>
> (3) The authors should explicitly discuss the potential limitations of the dataset, including possible biases in the data collection process or inherent limitations due to the specific domain. The authors may want to discuss how the dataset can be used responsibly to ensure fairness and privacy.
>
> Thanks for this comments. Following your advice, in the final version, we will add the following contents to discuss these aspects:
>
> 1. **Potential Biases Due to Labeler Backgrounds:** Our original intention in recruiting labelers from various backgrounds is to enhance the data diversity and avoid the data being affected by a single group. In the next version, we will discuss the steps we've taken to mitigate biases, such as providing clear guidelines to labelers and conducting quality checks. And we will explore the strategies to enhance the dataset's fairness and representation across various user demographics.
>
> 2. **Generalization to Other Content Types:** While our initial focus has been on short video recommendations, we acknowledge the need to explore the adaptability of our approach to different content types. We will provide insights into how our approach can be extended to other kinds of scenarios, such as news recommendation and book recommendation, by adjusting the annotation process and ensuring that the principles of explanation remain applicable.
>
> 3. **Responsible Usage:** We will outline measures taken to ensure fairness in data collection and annotation processes, as well as privacy protection mechanisms employed to safeguard user information such as data anonymization techniques and compliance with data protection regulations. Additionally, we will provide guidelines for responsible usage of the dataset, emphasizing the need for transparency, accountability, and adherence to ethical guidelines in research community.
>
> (4) The authors should consider discussing how the subjectivity of explanation satisfaction could potentially influence the model's evaluation.
>
> Thanks for this comments. We believe in the study of user subjective decision-making processes (e.g., explainable recommendation), the inherent biases and inconsistencies are indeed inevitable. We have carefully considered these aspects. To reduce such negative impact as much as possible, we have tried our best to design different strategies to control the quality (see section 3.2.2).
>
> (5) They should also discuss any potential misuse of the dataset and how such issues could be mitigated.
>
> Thanks for your comments. We have indeed given careful thought to potential misuse in the development of our dataset, which includes the following aspects:
>
> 1. **Privacy Protection:** We have adopted some approaches to protect privacy as much as possible, including data anonymization techniques and compliance with data protection regulations.
> 2. **Responsible Use:** It is important to stress the importance of using the dataset responsibly and in alignment with ethical guidelines. We will provide a detailed guidance on how researchers and should approach the dataset to prevent misuse.
> 3. **Mitigation Strategies:** We will discuss the strategies and mechanisms to mitigate potential misuse. This may include access controls, restricted use cases, and guidelines for sharing results derived from the dataset.
> 4. **Transparency and Accountability:** We will emphasize the need for transparency in research and the importance of being responsible for the implications of one's work.
>
> By discussing these aspects comprehensively, we aim to the responsible usage within the research community.
>
> Again, thanks for your comments. We believe most of the concerns are around the experiments on our dataset and its potential impacts. We really believe these aspects can be easily addressed in our final version, which should not influence the main contributions of our dataset too much.

---

> > ### Author Response · Authors · 2023-08-18
> > **Thanks for the comments**
> >
> > Dear reviewer 1MRi,
> >
> > Thanks again for your significant comments, which can definitely improve our paper. In the rebuttal and re-submitted paper, we have followed your advice to make more experiments on the error bar and incorporate the dataset statistics in the resubmitted paper. In addition, we will definitely add more clarifications on the limitations of our paper.
> >
> > We believe the above points can be easily addressed in our final version.
> >
> > Thanks

---

> > > ### Comment · Reviewer_1MRi · 2023-08-27
> > >
> > > Thank you for addressing my concerns in the rebuttal. I am satisfied with the explanations provided and am willing to revise my score upward for the manuscript.

---

> > > > ### Author Response · Authors · 2023-08-27
> > > > **Thanks for the response**
> > > >
> > > > Thanks so much for your responses, and in the final version, we will definitely revise our paper following your advice.

---

### Official Review · Reviewer_zpBA · 2023-07-21
**Review:  paper 312 REASONER**

**Rating:** 6
**Confidence:** 4
**Correctness:** Yes, the provided dataset is construc…
**Clarity:** Yes, it is easy to follow.

**Strengths:**

This dataset is constructed by about 3000 real users with different backgrounds, which contains
(i) Comprehensive user information: this dataset has collected user profiles like age, gender, income, hobby and physiology information, i.e. personalities.
(ii) Multi-aspect Explanation Ground Truth: this dataset provides the explanation ground truths from the perspectives of persuasiveness, informativeness and satisfaction.
(iii) Multi-modal Explanations:  in the dataset, users can select both textual and visual explanations according to their preferences for each video.
This dataset will make significant contribution to recommendation system research community, including explainable recommendation, sentiment analysis and sequential recommendation.


**Additional Feedback:**

(1) Please carefully check the submission to remove grammatical and typographical errors. For example:
     line 10 “explaination” should be “explanation”
    line 183 “our dataset may also contribute the field of debiased recommendation” -->“ contribute to the field.”
    line 190 “which make her do not want to” --> “ which make her not want to”
    ines 280, 282, 285 and 301, “persona”--> “personal”.

(2) line 114,  “All the videos are from an industrial short-video website”. It is better to add an URL or reference here.

(3) In the Appendix, both SOTA algorithms and representative explainable recommender models should be included in the benchmark. But in the submission, the latest algorithm was published in 2021.


**Documentation:**

Yes

**Limitations:**

Yes.

**Opportunities For Improvement:**

1. Quantity and data diversity can be further improved.
2. More SOTA explainable recommendation models should be included in the benchmark.


**Relation To Prior Work:**

Yes

**Summary And Contributions:**

This submission constructs a new explainable recommendation dataset and provide extensive analysis on its characteristics. Contributions include (i) multi-modal and multi-aspect explanation ground truths are labelled by about 3000 real users; (ii) a library is developed, where ten well-known explainable recommender models are implemented in a unified framework; and (iii) several benchmarks are built based on the proposed library for different explainable recommendation tasks.

---

> ### Author Response · Authors · 2023-08-16
>
> Thanks for the comments. In the following, we try to alleviate your concern one by one.
>
> (1) Quantity and data diversity can be further improved. More SOTA explainable recommendation models should be included in the benchmark.
>
> Thanks for these comments. We appreciate your reviews regarding the potential for further improvement in terms of quantity and data diversity. In the next version, we plan to take the following steps to improve these two aspects.
>
> 1. **Data Collection Scaling:** We will focus on expanding the scale of data collection efforts, encompassing a broader range of short video recommendations. This includes sourcing content from diverse sources and considering various recommendation scenarios to ensure a more comprehensive dataset.
> 2. **Combination with Large Language Models (LLMs)**: A promising direction is to designing instructions and prompts based on our currently labeled dataset for instruction fine-tunning and prompt tunning. By harnessing the language capabilities of LLMs, we can effectively generate more comprehensive and contextually relevant explanations for the recommended videos. This approach can not only expand the size of our dataset but also greatly reduce the cost of manual labeling.
> 3. **Inclusion of Various User Profiles:** To promote diversity, we intend to include a wider array of user profiles, encompassing different demographics, preferences, and behaviors. This will enrich the dataset with a more holistic representation of user interactions.
> 4. **Contextual Variation:** Recognizing the significance of context in recommendation systems, we plan to introduce more diverse contextual information, such as timestamp, geographical location and device type. This helps to improve the diversity of the dataset and make it closer to real recommendation scenarios
>
> By implementing this measures, we aim to achieve a significant improvement in terms of quantity and diversity.
>
>
> For the explainable recommendation models, **we have added seven recent models[1-7] for recommendation explanations (in the form of tags) in the resubmitted Appendix**.
>
> [1] Yi Zuo, et al. A tag-ware recommendation algorithm based on deep learning and multi-objective optimization. PRMVIA 2023.
>
> [2] Yi Zuo, TRAL: A Tag-Aware Recommendation Algorithm Based on Attention Learning. Applied Sciences 2023.
>
> [3] Weibin Zhao, et al. Hyperbolic Personalized Tag Recommendation. DASFAA 2022.
>
> [4] Bo Chen, et al. AIRec: Attentive intersection model for tag-aware recommendation. Neurocomputing 2021.
>
> [5] Jianshan Sun, et al. Hierarchical attention model for personalized tag recommendation. JASIST 2020
>
> [6] Ruoran Huang, et al. TNAM: A tag-aware neural attention model for Top-Nrecommendation. Neurocomputing 2020.
>
> [7] Hongmei Li, et al. Tag-aware recommendation based on Bayesian personalized ranking and feature mapping. Intelligent Data Analysis 2019.
>
>
>
>
> (2) Please carefully check the submission to remove grammatical and typographical errors. For example: line 10 “explaination” should be “explanation” line 183 “our dataset may also contribute the field of debiased recommendation” -->“ contribute to the field.” line 190 “which make her do not want to” --> “ which make her not want to” ines 280, 282, 285 and 301, “persona”--> “personal”.
>
> Thanks for pointing out these typo errors. We have revised them in the latest version.
>
>
>
> (3) line 114, “All the videos are from an industrial short-video website”. It is better to add an URL or reference here. In the Appendix, both SOTA algorithms and representative explainable recommender models should be included in the benchmark. But in the submission, the latest algorithm was published in 2021.
>
> Thanks for these comments. In the final version, we will add the website link (https://www.bilibili.com).
>
> For the benchmark, we have added seven recent models in the library (see the answer to (1) and the resubmitted paper).

---

> > ### Author Response · Authors · 2023-08-18
> > **Thanks for the comments**
> >
> > Dear reviewer zpBA,
> >
> > Thanks again for your significant comments, which can definitely improve our paper. In the rebuttal and re-submitted paper, we try to explain your questions one by one. In addition, we have added a large number of experiments to make our paper more solid. If you have further questions, we are very happy to discuss them.
> >
> > Thanks

---

> > > ### Author Response · Authors · 2023-08-29
> > > **Thanks for the comments**
> > >
> > > Dear Reviewer zpBA,
> > >
> > > Thanks so much for your hard work in reading and reviewing our paper.
> > >
> > > Considering that the rebuttal ddl is approaching rapidly, we would like to kindly ask whether our rebuttal and additional experiments can alleviate your concerns. If you have more questions, we would like to grasp the last opportunities to discuss them for a higher score.
> > >
> > > Your input is highly valued and helps us improve our work.
> > >
> > > Thanks

---

### Official Review · Reviewer_VntB · 2023-07-23
**Review for An Explainable Recommendation Dataset with Comprehensive Labeling Ground Truths**

**Rating:** 6
**Confidence:** 5
**Correctness:** Yes. The whole collection process is …
**Clarity:** Yes.

**Strengths:**

The strengths are as follows:
- This dataset contains explanations for short videos from multiple aspects and comprehensive
- The persona information is available for  recommendation
- The data collection process is in details and clear
- This will motivate the current explainable recommendation research

**Additional Feedback:**

Another possible direction for using this new benchmark may be shifting the new benchmark as the instructions or prompts for large pre-trained models. This will enable existing large pre-trained models with better explanation ability for short videos.

**Documentation:**

The dataset is released and in good documentation.

**Ethics:**

The video may contain sensitive information, which requires the authors to double check.

The data collection process is well explained and has no ethical concerns.

**Limitations:**

- As this paper already mentioned in their conclusion section, though the explanation information is comprehensive, however, the data scale is limited. This dataset only has 2,997 users, 4,672 items, and 6,115 tags. If considering the user-item interaction, it only has 58,497, which is much fewer than some existing small recommendation dataset benchmark, such as  MovieLens 100K. Thus, it may have limited contribution. It would

- This datasets may not be aligned with real-world recommender systems. Most of the short video recommendation factors are not considered in the explanation such as the trending topics, the fashion styles, the dwelling times, the session behavior, the time periods, and etc. It would be better to collect the dataset from real user usage scenario rather than ask labeler to watch the video and collect the feedback. Some complex context information may also be important for explaining the short video watching behaviors of users.

- No baselines are compared for this benchmark. It would be better to see how existing SOTA RecSys methods perform on this new benchmark.




**Opportunities For Improvement:**

- It would be better to compare the performance of different baselines on this new benchmark, such as video recommendation, multi-aspect recommendation, explainability in recommendation, etc.

- For the explanation question Q123. For example, Q2 Which features are most informative for this video? I am wondering what are those features extracted? In Figure 1(a), all Q123 are int set answers. How is the int set constructed?

- Also, for video recommendation, especially the short video recommendation, it would be better to investigate the dwelling time of users. In modern short video platform, like Douyin/TikTok or Instagram short video, users may not complete the watch of short video if they feel dislike the content.

**Relation To Prior Work:**

Yes.

**Summary And Contributions:**

This paper describe a new dataset for the explainable recommendation, especially for short video recommendation as the items are from an industrial short-video website.

- The dataset collection procedure is well presented and explainable, which will benefit the follow-up works to use this datasets.
- Additionally, the explanation of why user watch the video is in comprehensive details. This will promote existing explainable recommendation research that only use reviews as explanation.
- Moreover, extensive persona information is also available, which will motivate more specific personalized video recommendation research.

---

> ### Author Response · Authors · 2023-08-16
>
> Thanks for the comments. In the following, we try to alleviate your concern one by one.
>
> (1) It would be better to compare the performance of different baselines on this new benchmark.
>
> Actually, we have made such comparisons, which are presented in the Appendix. In the Appendix, we have built a library, and compared all the baselines in the library in terms of video recommendation (*i.e.* the task of rating prediction) , multi-aspect recommendation (*i.e.* the task of tag prediction) and explainable recommendation (*i.e.* the task of natural language explanations generation) based on our datasets.
> In addition to existing models presented in the Appendix, **we also added seven recent models[1-7] for recommendation explanations (in the form of tags) in the resubmitted Appendix**. We hope that our efforts can alleviate your concerns.
>
> [1] Yi Zuo, et al. A tag-ware recommendation algorithm based on deep learning and multi-objective optimization. PRMVIA 2023.
>
> [2] Yi Zuo, TRAL: A Tag-Aware Recommendation Algorithm Based on Attention Learning. Applied Sciences 2023.
>
> [3] Weibin Zhao, et al. Hyperbolic Personalized Tag Recommendation. DASFAA 2022.
>
> [4] Bo Chen, et al. AIRec: Attentive intersection model for tag-aware recommendation. Neurocomputing 2021.
>
> [5] Jianshan Sun, et al. Hierarchical attention model for personalized tag recommendation. JASIST 2020
>
> [6] Ruoran Huang, et al. TNAM: A tag-aware neural attention model for Top-Nrecommendation. Neurocomputing 2020.
>
> [7] Hongmei Li, et al. Tag-aware recommendation based on Bayesian personalized ranking and feature mapping. Intelligent Data Analysis 2019.
>
> (2) For the explanation question Q123. For example, Q2 Which features are most informative for this video? I am wondering what are those features extracted? In Figure 1(a), all Q123 are int set answers. How is the int set constructed?
>
> 1. For each video, we present four types of features to the users, that is, the title, the brief introduction, the tags and the preview information (see step 2 of Figure 2).
> 2. For easy operation, we first project the tags into IDs, and then introduce 7 special IDs indicating the title, brief introduction and five previews. The tag IDs range from 0 to 6114, and the IDs for the title, brief introduction and five previews are from 6115 to 6121.
> 3. We use the special tags (6115 to 6121) for unifying the record format, and one can easily retrieve the real contents of the title, brief introduction and previews based on the current video ID. **For the tag-ID projection dict, it has been included in the dataset**.
> 4. The answering sets for Q1, Q2 and Q3 are the IDs corresponding to the user selected features.
>
> (3) For video recommendation, it would be better to investigate the dwelling time of users. In short video platform, users may not complete the watch of short video if they feel dislike the content.
>
> Since we aim to label the explanation ground truth, we would like to ensure that the users have watched the complete video, so that they can correctly express their feelings based on the video contents. We believe investigating the dwelling time of users is very interesting, which can be left for future work.
>
> (4) As this paper already mentioned in conclusion section, though the explanation information is comprehensive, however, the data scale is limited. If considering the user-item interaction, it only has 58,497, which is much fewer than some existing small recommendation dataset benchmark, such as MovieLens 100K. Thus, it may have limited contribution.
>
> Thanks for this comment. The meaning of this dataset is to make up the gap that most public datasets do not have explanation ground truth. In the field of explainable recommendation, our dataset is already 10 times larger than the ones which are usually leveraged to evaluate the explainable models (see table 1). While the scale of our dataset is not that large as the ones without explanation ground truth, it can provide many new opportunities which has been ignored by the existing datasets. We believe the contributions are non-trivial.
>
> (5) This datasets may not be aligned with real-world recommender systems. It would be better to collect the dataset from real user usage scenario rather than ask labeler to watch the video and collect the feedback. Some complex context information may also be important for explaining the short video watching behaviors of users.
>
> Thanks for this comment. We believe this suggestion is very interesting. In the next version of our dataset, we will definitely follow your advice to collect the dataset in real applications.
>
> (6) Another possible direction for using this new benchmark may be shifting the new benchmark as the instructions or prompts for large pre-trained models. This will enable existing large pre-trained models with better explanation ability.
>
> This is a very interesting idea, thanks for this suggestion. We will try to combine our dataset with large language models to extend its impacts.

---

> > ### Author Response · Authors · 2023-08-18
> > **Thanks for the comments**
> >
> > Dear reviewer VntB,
> >
> > Thanks again for your significant comments, which can definitely improve our paper. In the rebuttal and re-submitted paper, we try to explain your questions one by one. In addition, we have added a large number of experiments to make our paper more solid. If you have further questions, we are very happy to discuss them.
> >
> > Thanks

---

> > > ### Comment · Reviewer_VntB · 2023-08-21
> > > **Reply to author comments**
> > >
> > > Thanks for answering my questions and this new results are good to support the benefits of this dataset. I would keep my score and pretty lien to accept this paper.

---

> > > > ### Author Response · Authors · 2023-08-21
> > > > **Thanks for the comments**
> > > >
> > > > Thanks so much for your initial insightful comments and the prompt responses to our rebuttal.

---

### Author Response · Authors · 2023-08-26
**Author Official Comment**

Dear Reviewers,

Thanks so much for your hard work in reading and reviewing our paper.  All your comments are really helpful to improve our paper.
We greatly appreciate all feedback and are committed to participating in ongoing discussions, particularly when reviewers present comments or questions. Your input is highly valued and helps us improve our work.

Thanks

---

### Decision · Program_Chairs · 2023-09-22

**Decision:**

Accept (Poster)

**Comment:**

The authors present a new dataset for explainable recommendations, based on multi-modal short video recommendations, and they give extensive analysis of the dataset.

The reviewers generally agreed that this paper gives a substantial contribution, especially because it overcomes limitations of existing datasets. Furthermore, the reviewers appreciated the strong methodology and execution of the paper.

The reviewers noted weaknesses such as the limited data scale, potentially limited alignment with real-world recommender systems, and data diversity. The authors put in significant effort during the discussion period to improve the paper, and in the end all reviewers were positive about the paper.

Overall, the paper is a strong contribution, and I recommend acceptance.